# A genome-wide meta-analysis yields 46 new loci associating with biomarkers of iron homeostasis

Steven Bell [1,2,35], Andreas S. Rigas[3,35], Magnus K. Magnusson [4,5,35✉], Egil Ferkingstad [4,35], Elias Allara [1,2,35], Gyda Bjornsdottir[4], Anna Ramond[1,2,6], Erik Sørensen[3], Gisli H. Halldorsson [4], Dirk S. Paul [1,2], Kristoffer S. Burgdorf[3], Hannes P. Eggertsson [4], Joanna M. M. Howson [2], Lise W. Thørner[3], Snaedis Kristmundsdottir[4], William J. Astle[1,2,7,8], Christian Erikstrup [9], Jon K. Sigurdsson[4], Dragana Vuckovic[1,8], Khoa M. Dinh[9], Vinicius Tragante [4,10], Praveen Surendran[2,11], Ole B. Pedersen [12], Brynjar Vidarsson[13], Tao Jiang[1,2,8], Helene M. Paarup [14], Pall T. Onundarson[5,15], Parsa Akbari [1,2,8], Kaspar R. Nielsen[16], Sigrun H. Lund [4], Kristinn Juliusson[4], Magnus I. Magnusson[4], Michael L. Frigge [4], Asmundur Oddsson [4], Isleifur Olafsson[17], Stephen Kaptoge[1,2], Henrik Hjalgrim[18], Gudmundur Runarsson[13], Angela M. Wood[1,2], Ingileif Jonsdottir [4,5], Thomas F. Hansen [19,20,21], Olof Sigurdardottir[22], Hreinn Stefansson [4], David Rye[23], DBDS Genomic Consortium*, James E. Peters[2], David Westergaard [24], Hilma Holm [4], Nicole Soranzo [1,8,25], Karina Banasik [24], Gudmar Thorleifsson[4], Willem H. Ouwehand [1,8,25,26], Unnur Thorsteinsdottir[4,5], David J. Roberts[1,27,28], Patrick Sulem [4], Adam S. Butterworth [1,2], Daniel F. Gudbjartsson [4,29], John Danesh[1,2,8], Søren Brunak [24], Emanuele Di Angelantonio[1,2,26,36✉], Henrik Ullum[3,36✉] & Kari Stefansson [4,5,36✉]

Iron is essential for many biological functions and iron deficiency and overload have major health implications. We performed a meta-analysis of three genome-wide association studies from Iceland, the UK and Denmark of blood levels of ferritin ($N = 246,139$), total iron binding capacity ($N = 135,430$), iron ($N = 163,511$) and transferrin saturation ($N = 131,471$). We found 62 independent sequence variants associating with iron homeostasis parameters at 56 loci, including 46 novel loci. Variants at *DUOX2*, *F5*, *SLC11A2* and *TMPRSS6* associate with iron deficiency anemia, while variants at *TF*, *HFE*, *TFR2* and *TMPRSS6* associate with iron overload. A *HBS1L-MYB* intergenic region variant associates both with increased risk of iron overload and reduced risk of iron deficiency anemia. The *DUOX2* missense variant is present in 14% of the population, associates with all iron homeostasis biomarkers, and increases the risk of iron deficiency anemia by 29%. The associations implicate proteins contributing to the main physiological processes involved in iron homeostasis: iron sensing and storage, inflammation, absorption of iron from the gut, iron recycling, erythropoiesis and bleeding/menstruation.

A full list of author affiliations appears at the end of the paper.

ron is an essential element for a wide variety of metabolic processes such as oxygen transport, cellular respiration, and redox reactions in numerous metabolic pathways. For this reason, iron homeostasis is tightly regulated on cellular and systemic levels to ensure a balance between uptake, transport, storage, and utilization. Iron deficiency is one of the five leading causes of disability worldwide, especially among children and women of childbearing age[1,2]. Similarly, iron overload is associated with an increased risk of several major chronic conditions, including diabetes and liver disease[1,3].

Four iron biomarkers are used for clinical assessment of iron status: serum ferritin, serum iron, and total iron-binding capacity (TIBC) are measured directly, while transferrin saturation (TSAT) is derived as serum iron divided by TIBC. While serum ferritin correlates well with body iron stores in non-inflamed individuals[4], TSAT measures the proportion of iron-binding sites of transferrin that are occupied by iron. TSAT indicates the availability of iron for erythropoiesis and is low in iron deficiency and high during iron overload. In some forms of anemia (e.g., anemia of inflammation) the iron is not transported efficiently to the bone marrow for erythropoiesis, despite adequate iron stores. Since in this situation there is adequate ferritin but low TSAT, it is useful to evaluate TSAT in addition to ferritin[4,5].

Genome-wide association studies (GWAS) have previously investigated the association between sequence variants and iron homeostasis biomarkers[6–8]. The largest study to date yielded 11 loci: *ABO*, *ARNTL*, *FADS2*, *HFE*, *NAT2*, *SLC40A1*, *TEX14*, *TF*, *TFR2*, *TFRC*, and *TMPRSS6* associating with one or more iron homeostasis biomarkers (ferritin, iron, TIBC or TSAT)[6]. To search for additional sequence variants associated with iron homeostasis, we performed a GWAS meta-analysis of ferritin, serum iron, TIBC, and TSAT in Iceland and blood donor studies from the UK (INTERVAL study) and Denmark (Danish Blood Donor Study). This was followed by cross-referencing of iron-associated loci with clinically relevant phenotypes (including iron deficiency anemia (IDA), iron overload, and red blood cell indices). We report associations with iron homeostasis biomarkers for 62 independent sequence variants at 56 loci, including 46 novel loci. Based on a literature review, we categorize 25 of these loci as involved in iron sensing or storage, inflammation, gut absorption, iron recycling, erythropoiesis, and bleeding/menstruation.

## Results

**Overview.** We performed a meta-analysis of four iron-related biomarkers: ferritin ($N = 246{,}139$), serum iron ($N = 163{,}511$), TIBC ($N = 135{,}430$), and TSAT ($N = 131{,}471$), combining GWAS results from Iceland, the UK, and Denmark (Fig. 1, Supplementary Data 1). We found associations with iron homeostasis biomarkers represented by 62 sequence variants at 56 loci, of which 46 have not been reported in the previous GWAS on iron homeostasis and are therefore considered novel (Table 1, Table 2, Fig. 2, and Supplementary Data 2). For each locus, we report the lead variant (lowest $P$ value) and additional uncorrelated variants ($r^2 < 0.1$) within the locus with genome-wide significance. Our criteria for statistical significance have been previously described[9] (see "Methods"). A variant-to-gene mapping algorithm that takes into account gene location, variant effect (for coding variants), and effect on gene expression (eQTL) for each variant (lead variant and LD class) was used to choose a single candidate gene for each locus (see "Methods"). Twenty-five of the 62 iron homeostasis-associated sequence variants have a high-confidence predicted causal gene, 23 variants have multiple top-scoring genes, 36 variants have at least one coding variant or eQTL in the LD class, and 13 variants have more than one gene with coding variants and/or eQTL in the LD class (Supplementary Data 3). The LD class of a variant is defined as all variants having $r^2 > 0.8$ with the variant. Linkage disequilibrium (LD) ($r^2$) is estimated based on the Icelandic population. In cases where variants had more than one top-scoring gene, the gene closest to the lead variant was selected, except for two loci where likely candidate genes were present among the top-scoring genes (*FTL* (ferritin light chain) and *HAMP* (hepcidin)) (Supplementary Data 3). Fourteen of the variants associated with more than one biomarker, bringing the total number of observed associations to 87 (Supplementary Data 2). All our associations have $P < 3.0 \times 10^{-8}$. We replicated the association of all 11 previously reported variants[6], 10 at genome-wide significance (Supplementary Data 2). In addition, we found six rare variants (minor allele frequency (MAF) < 1%), six low-frequency variants (1% ≤ MAF < 5%), and 37 common variants that have previously not been reported to associate with iron homeostasis biomarkers (Supplementary Data 2). Forty-six variants associate with a single iron biomarker (ferritin, 34;

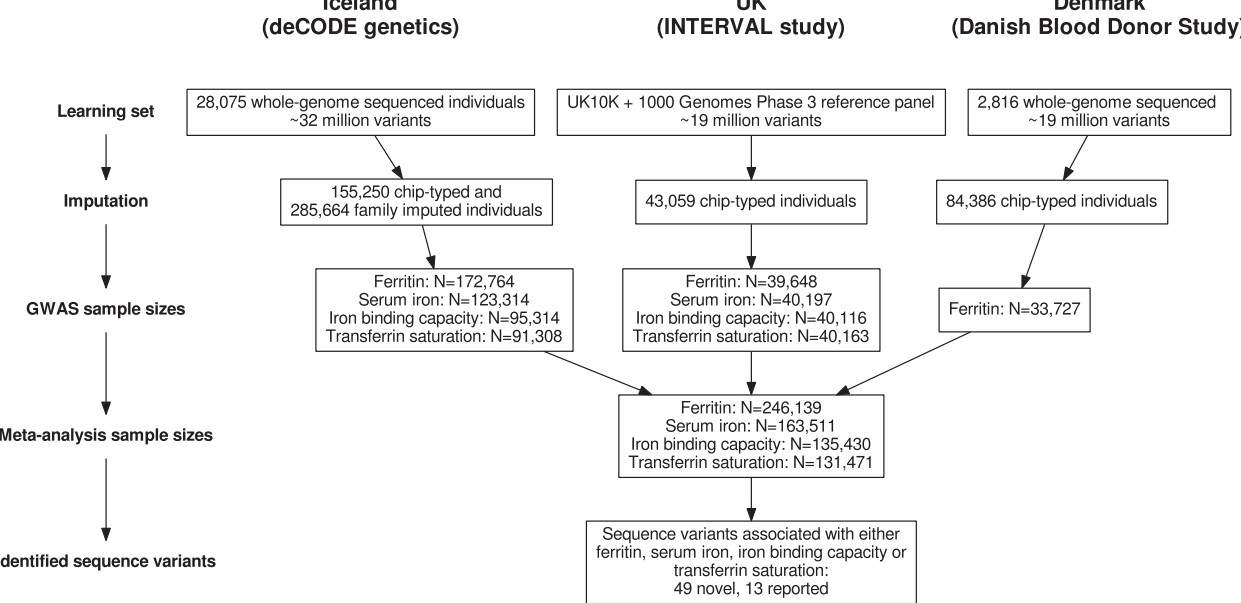

**Fig. 1 Study design for meta-GWAS of iron homeostasis biomarkers.** A flowchart describing the study design of the iron homeostasis biomarker GWAS meta-analysis of Icelandic, UK, and Danish data.

**Table. 1 Novel iron homeostasis associated variants found in the meta-analysis of Icelandic, UK, and Danish data, excluding variants that are only associated with ferritin.**

| Marker | Position (hg38) | MAF (%) | Min/maj | Gene | Phenotype | The effect in SD (95% CI) | P value | $P_{het}$ |
|---|---|---|---|---|---|---|---|---|
| rs35945185 | chr1:65671556 | 36.5 | A/G | LEPR[a] | Iron | 0.031 (0.023, 0.039) | $1.54 \times 10^{-13}$ | 0.00130 |
| rs469882 | chr1:91064875 | 20.8 | C/A | ZNF644 | TIBC | −0.037 (−0.048, −0.026) | $1.03 \times 10^{-10}$ | 0.566 |
| rs2228145 | chr1:154454494 | 40.7 | C/A | IL6R[a] | Iron | 0.026 (0.018, 0.034) | $8.42 \times 10^{-11}$ | 0.107 |
| rs6025 | chr1:169549811 | 2.79 | T/C | F5[a] | Ferritin | 0.15 (0.12, 0.17) | $6.93 \times 10^{-37}$ | $7.90 \times 10^{-9}$ |
| | | | | | TIBC | −0.093 (−0.13, −0.061) | $1.76 \times 10^{-8}$ | 0.227 |
| rs13007705 | chr2:238160555 | 42.5 | T/C | ERFE | Iron | 0.029 (0.021, 0.037) | $2.01 \times 10^{-12}$ | 0.717 |
| | | | | | TSAT | 0.033 (0.024, 0.042) | $1.10 \times 10^{-12}$ | 0.0539 |
| rs7630745 | chr3:66376605 | 35.8 | C/T | LRIG3[a] | Iron | 0.025 (0.017, 0.033) | $2.09 \times 10^{-9}$ | 0.924 |
| rs59950280 | chr4:3450618 | 33.5 | A/G | HGFAC[a] | TIBC | 0.033 (0.023, 0.043) | $5.87 \times 10^{-11}$ | 0.729 |
| rs9399136 | chr6:135081201 | 25.9 | C/T | MYB[a] | Iron | 0.057 (0.049, 0.066) | $1.08 \times 10^{-36}$ | 0.492 |
| | | | | | TIBC | −0.033 (−0.044, −0.023) | $2.65 \times 10^{-10}$ | 0.483 |
| | | | | | TSAT | 0.067 (0.057, 0.077) | $5.32 \times 10^{-39}$ | 0.130 |
| rs12718598 | chr7:50360747 | 46.3 | C/T | IKZF1 | Iron | 0.027 (0.019, 0.034) | $3.69 \times 10^{-11}$ | 0.126 |
| rs17580 | chr14:94380925 | 3.79 | A/T | SERPINA1[a] | TIBC | 0.076 (0.053, 0.099) | $1.19 \times 10^{-10}$ | 0.811 |
| rs57659670 | chr15:45106240 | 7.53 | C/T | DUOX2[a] | Ferritin | −0.14 (−0.16, −0.13) | $1.05 \times 10^{-113}$ | 0.0361 |
| | | | | | Iron | −0.042 (−0.056, −0.028) | $1.08 \times 10^{-8}$ | 0.0708 |
| | | | | | TSAT | −0.058 (−0.074, −0.041) | $5.73 \times 10^{-12}$ | 0.876 |
| | | | | | TIBC | 0.077 (0.06, 0.094) | $3.67 \times 10^{-19}$ | 0.434 |
| rs77262773 | chr17:69253570 | 2.61 | T/C | ABCA5 | Iron | 0.081 (0.055, 0.11) | $9.54 \times 10^{-10}$ | 0.130 |
| rs2005682 | chr19:35456759 | 30.5 | T/A | HAMP | TSAT | −0.032 (−0.042, −0.022) | $6.25 \times 10^{-11}$ | 0.126 |
| | | | | | Iron | −0.029 (−0.037, −0.02) | $2.37 \times 10^{-11}$ | 0.270 |
| rs112727702 | chr19:49587947 | 23.2 | T/G | PRRG2 | TIBC | 0.043 (0.032, 0.054) | $2.08 \times 10^{-14}$ | 0.849 |
| rs1132274 | chr20:17615510 | 16.8 | A/C | RRBP1[a] | TIBC | 0.036 (0.023, 0.048) | $1.93 \times 10^{-8}$ | 0.0837 |

[a]High-confidence predicted causal gene (based on a variant-to-gene algorithm, see "Methods").
Min/maj minor/major allele, MAF minor allele frequency, Gene predicted causal gene based on a variant-to-gene algorithm (see "Methods"), SD standard deviation, CI confidence interval, $P_{het}$ P value from the test for heterogeneity (see "Methods"), TIBC total iron-binding capacity, TSAT transferrin saturation. The effect is shown for the minor allele.

iron, 8; TIBC, 4), while 12 variants associate with more than one biomarker (Fig. 3). Only three variants associate with all four iron biomarkers: p.His678Arg in *DUOX2* (rs57659670[C]; MAF = 7.53%), p.Cys282Tyr in *HFE* (rs1800562[A]; MAF = 6.77%), and p.Val749Ala in *TMPRSS6* (rs855791[A]; MAF = 43.1%). The missense variant at *DUOX2*, a dual oxidase involved in the generation of $H_2O_2$[10], has not previously been associated with iron homeostasis.

We calculated the correlation between the iron biomarkers and selected other biomarkers related to iron metabolism (red blood cell indices, platelet count, erythrocyte sedimentation rate, and C-reactive protein) (Supplementary Fig. 1) and the genetic correlation between the four iron biomarkers (Supplementary Data 4). Among the iron homeostasis biomarkers, the strongest correlation was between iron and TSAT (0.86) and the strongest genetic correlation was also between these biomarkers (Iceland TSAT vs. UK iron: 0.53 (SE = 0.19), $P = 0.0059$; Iceland iron vs. UK TSAT: 0.54 (SE = 0.17), $P = 0.0020$) (Supplementary Data 4). Furthermore, we estimated the heritability of the iron homeostasis biomarkers to be between 0.16 and 0.32 using parent–offspring and sibling correlations, suggesting that heritability explains 16–32% of the variance of the four iron homeostasis markers studied (Supplementary Data 5).

We tested for heterogeneity between the results from the Icelandic, UK, and Danish cohorts (Supplementary Data 2). Of the 87 associations, 79 are with markers present in two or more populations and of these, 19 show nominally significant heterogeneity ($P < 0.05$). For all associations, the effects are in the same direction in all three populations, and for 79 of the 87 associations, effects are nominally significant ($P < 0.05$) in all three populations (Supplementary Fig. 2, Supplementary Fig. 3). Eight associations are reported with five rare variants at three loci found only in Iceland (MAF = 0.12–0.47%): three coding variants (two missense, one stop-gained) in *STAB1*, a stop-gained variant in *TF*, and a stop-gained variant in *TMPRSS6*. Common variants associating with iron biomarkers are reported in all three populations for each of these loci, providing additional evidence for these associations (Supplementary Data 2).

Because of the well-known difference in iron homeostasis between the sexes[11], we tested for sexual dimorphism in iron biomarker associations (Supplementary Data 6). We found differences in the ferritin effect (using a test for heterogeneity with P value threshold $P < 0.05/62 = 8.1 \times 10^{-4}$) of five of the 62 variants. In addition, we identified one additional variant that only associates with ferritin in women: a missense variant in *VWF* (p.Tyr1584Cys/rs1800386[C]), a likely pathogenic type 2 von Willebrand disease (VWD) mutation[12] ($\beta = -0.17$ standard deviation (SD) [−0.23, −0.12], $P = 3.0 \times 10^{-10}$). Of the six variants, four have greater effects in women (*F5*: six times greater effect, *SLC25A37*: three times greater effect, *DUOX2*: 36% greater effect, and *VWF*: 13 times greater effect) and two in men (*HK1*: four times greater effect, *HFE* p.Cys282Tyr: 51% greater effect) (Supplementary Data 6). The four variants with larger effects in women also have stronger effects in premenopausal than postmenopausal women (Supplementary Data 7). In addition, we find sex differences in variants in the well-known iron regulatory genes *HFE* (ferritin, iron, and TSAT) and *TMPRSS6* (iron) (Supplementary Data 6), again with stronger effects in pre- vs. postmenopausal women (Supplementary Data 7). For the variants at *F5*, *SLC25A37*, *DUOX2*, and *VWF* that show a greater effect on women, the difference does not persist when comparing only men and postmenopausal women (Supplementary Data 8).

**Iron homeostasis variants and protein quantitative loci (pQTL).** To gain further insight into the biological pathways involved in iron homeostasis, we tested for association of the

**Table. 2 Novel ferritin-associated variants found in the meta-analysis of Icelandic, UK, and Danish data, excluding variants that also associate with other iron homeostasis biomarkers (iron, TIBC, and TSAT).**

| Marker | Position (hg38) | Min/maj | MAF (%) | Gene | The effect in SD (95% CI) | P value | P_het |
|---|---|---|---|---|---|---|---|
| rs75965181 | chr1:22257509 | A/T | 2.14 | WNT4 | −0.12 (−0.14, −0.097) | $3.70 \times 10^{-26}$ | 0.709 |
| rs10801913 | chr1:115671658 | A/G | 30.7 | VANGL1[a] | 0.024 (0.016, 0.031) | $2.63 \times 10^{-10}$ | 0.0243 |
| rs551459670 | chr1:220115348 | A/G | 1.10 | IARS2[a] | 0.14 (0.1, 0.18) | $1.28 \times 10^{-13}$ | 0.141 |
| rs1260326 | chr2:27508073 | T/C | 36.8 | GCKR | 0.025 (0.018, 0.032) | $1.48 \times 10^{-12}$ | 0.0035 |
| rs6757653 | chr2:28948938 | T/C | 27.4 | WDR43[a] | 0.032 (0.024, 0.039) | $9.34 \times 10^{-16}$ | 0.343 |
| rs1250259 | chr2:215435759 | T/A | 28.8 | FN1[a] | −0.024 (−0.032, −0.017) | $1.84 \times 10^{-10}$ | 0.459 |
| rs762752083 | chr3:52502023 | T/G | 0.24 | STAB1[a] | 0.35 (0.26, 0.44) | $3.19 \times 10^{-14}$ | – |
| rs750717575 | chr3:52502709 | A/G | 0.27 | STAB1[a] | 0.24 (0.16, 0.32) | $2.18 \times 10^{-8}$ | – |
| rs745795585 | chr3:52505379 | A/G | 0.47 | STAB1[a] | 0.29 (0.23, 0.35) | $2.60 \times 10^{-19}$ | – |
| rs34216132 | chr3:52693659 | C/G | 0.333 | STAB1[a] | 0.17 (0.11, 0.22) | $3.50 \times 10^{-9}$ | 0.443 |
| rs1131262 | chr3:134222476 | T/C | 11.2 | RYK[a] | −0.032 (−0.042, −0.021) | $6.66 \times 10^{-9}$ | 0.554 |
| rs36184164 | chr6:43813355 | G/T | 12.6 | VEGFA | 0.036 (0.025, 0.046) | $6.46 \times 10^{-12}$ | 0.358 |
| rs2529440 | chr7:30472178 | T/C | 44.6 | NOD1[a] | −0.035 (−0.041, −0.028) | $4.60 \times 10^{-23}$ | 0.0393 |
| rs4841429 | chr8:10711019 | G/A | 7.86 | RP1L1 | 0.06 (0.048, 0.073) | $8.21 \times 10^{-21}$ | 0.0893 |
| rs13253974 | chr8:23520397 | A/G | 32.3 | SLC25A37 | 0.024 (0.017, 0.032) | $2.51 \times 10^{-11}$ | $6.1 \times 10^{-9}$ |
| rs2954029 | chr8:125478730 | T/A | 47.9 | TRIB1 | −0.024 (−0.031, −0.018) | $1.42 \times 10^{-12}$ | 0.665 |
| rs7865362 | chr9:33117967 | T/C | 36.0 | B4GALT1 | 0.025 (0.018, 0.032) | $1.03 \times 10^{-11}$ | 0.223 |
| rs17476364 | chr10:69334748 | C/T | 10.8 | HK1 | 0.043 (0.032, 0.054) | $3.57 \times 10^{-14}$ | 0.123 |
| rs12419620 | chr11:2211323 | G/T | 16.1 | TH | −0.031 (−0.04, −0.022) | $3.43 \times 10^{-11}$ | 0.867 |
| rs12807014 | chr11:47738526 | C/T | 27.4 | FNBP4 | −0.029 (−0.036, −0.021) | $2.72 \times 10^{-13}$ | 0.246 |
| rs4938939 | chr11:60393365 | A/G | 29.3 | MS4A7[a] | 0.022 (0.015, 0.03) | $3.01 \times 10^{-9}$ | 0.33 |
| - | chr12:50983028 | b | 0.68 | SLC11A2 | −0.16 (−0.19, −0.13) | $1.46 \times 10^{-24}$ | $6.0 \times 10^{-5}$ |
| rs996347 | chr14:33941686 | C/T | 35.5 | EGLN3 | 0.049 (0.042, 0.056) | $2.99 \times 10^{-41}$ | 0.0656 |
| rs3743171 | chr15:65624189 | T/A | 19.1 | DPP8 | −0.024 (−0.032, −0.015) | $2.92 \times 10^{-8}$ | 0.644 |
| rs9921222 | chr16:325782 | C/T | 49.2 | AXIN1 | 0.025 (0.018, 0.032) | $1.09 \times 10^{-12}$ | 0.909 |
| rs3747602 | chr16:4752385 | G/T | 36.8 | ZNF500 | 0.021 (0.014, 0.028) | $2.47 \times 10^{-9}$ | 0.512 |
| rs535064984 | chr17:7116978 | C/T | 0.58 | ASGR1 | 0.23 (0.18, 0.28) | $3.61 \times 10^{-19}$ | 0.713 |
| rs55789050 | chr17:9890100 | T/A | 33.3 | GLP2R[a] | −0.027 (−0.034, −0.02) | $6.07 \times 10^{-14}$ | 0.768 |
| rs1542752 | chr17:74942005 | T/C | 15.3 | OTOP3[a] | 0.034 (0.025, 0.044) | $1.44 \times 10^{-12}$ | 0.0478 |
| rs708686 | chr19:5840608 | T/C | 25.8 | FUT6 | −0.031 (−0.039, −0.023) | $1.96 \times 10^{-14}$ | 0.317 |
| rs4808802 | chr19:18467063 | C/G | 21.8 | ELL | 0.028 (0.019, 0.036) | $3.42 \times 10^{-11}$ | 0.254 |
| rs601338 | chr19:48703417 | G/A | 48.4 | FUT2[a] | 0.028 (0.021, 0.035) | $7.04 \times 10^{-16}$ | 0.0019 |
| rs143041401 | chr19:49046859 | A/G | 1.61 | FTL | 0.11 (0.078, 0.13) | $4.03 \times 10^{-14}$ | 0.0017 |
| rs6029148 | chr20:40495768 | A/G | 7.10 | MAFB | 0.046 (0.033, 0.058) | $5.56 \times 10^{-12}$ | 0.036 |

Min/maj minor/major allele, MAF minor allele frequency, Gene predicted causal gene based on a variant-to-gene algorithm (see "Methods"), SD standard deviation, CI confidence interval, $P_{het}$ $P$ value from the test for heterogeneity (see "Methods"). The effect is shown for the minor allele.
[a]High-confidence predicted causal gene (based on a variant-to-gene algorithm, see "Methods").
[b]The minor allele is a 3.5 kb deletion in the 3′ UTR of SLC11A2.

62 iron homeostasis variants (including all variants with $r^2 \geq 0.8$ with any iron homeostasis variants) with an expression of 4792 proteins in serum using the SomaLogic Somascan platform based on samples from 35,559 Icelanders (Methods, Supplementary Data 9). Among the 62 variants, 30 have at least one associated pQTL, where we use $r^2 > 0.8$ as the limit for considering variants as associated. The variants at ABO, SERPINA1, FUT2, ABCA5, GCKR, and ASGR1 each have over 50 pQTL, with 24 other variants have between one and 14 pQTL. Interestingly, variants at or close to HFE (rs55925606[G], $\beta = -0.153$ SD [−0.186, −0.120], $P = 2.8 \times 10^{-20}$), MTMR4 (rs34523089[T], $\beta = 0.062$ SD [0.040, 0.084], $P = 2.3 \times 10^{-8}$) and TMPRSS6 (rs885791[A], $\beta = 0.060$ SD [0.044, 0.076], $P = 9.5 \times 10^{-13}$) all associate with protein levels of hepcidin. The variant at TMPRSS6 also associates with increased protein levels of erythropoietin ($\beta = 0.066$ SD [0.050, 0.082], $P = 1.6 \times 10^{-15}$) and transferrin receptor protein 1 ($\beta = 0.127$ [0.111, 0.143], $P = 1.7 \times 10^{-52}$). Furthermore, variants at LEPR and IL6R associate with decreased levels of the inflammatory mediator's serum amyloid A-1 and A-2 proteins, with the variant at LEPR also associating with reduced levels of C-reactive protein. The rs762752083[T] stop-gained variant at STAB1 associates with increased levels of von Willebrand factor ($\beta = 0.510$ SD [0.347, 0.673], $P = 7.9 \times 10^{-10}$). Finally, the rs199138[A] intron variant at DUOX2 associates with decreased levels of ferritin light chain ($\beta = -0.121$ SD [−0.150, −0.092], $P = 3.2 \times 10^{-16}$). This variant is in strong LD ($r^2 = 0.97$) with the DUOX2 His678Arg missense variant found to associate with a decrease in serum ferritin.

**The loci in the context of systemic iron homeostasis.** Based on a literature review, we placed 24 of the 56 candidate genes, as well as the female-specific candidate gene VWF, into 6 categories representing the main physiological processes involved in iron homeostasis: hepcidin regulation and iron storage (FTL, HAMP, HFE, TMPRSS6, TFR2, TFRC, TF, MTMR4, and SERPINA1), inflammation (IL6R, NOD1, and IKZF1), gut absorption (SLC11A2, SLC40A1, EGLN3, and DUOX2), iron recycling (SLC11A2, SLC40A1, STAB1, TRIB1, and MAFB), erythropoiesis (ERFE, SLC25A37, MYB, and HK1) and bleeding/menstruation (F5 and VWF) (Fig. 4).

*Hepcidin regulation and iron storage*: Synthesis of the iron homeostasis hormone hepcidin (HAMP) is under tight regulation by the liver iron sensing and signaling cascade involving several proteins, including those encoded by HFE, TMPRSS6, TF, TFR2, and TFRC[13]. Hepcidin as the major iron homeostasis hormone regulates iron transport from cells through inhibition (and degradation) of ferroportin in cells, such as intestinal epithelial

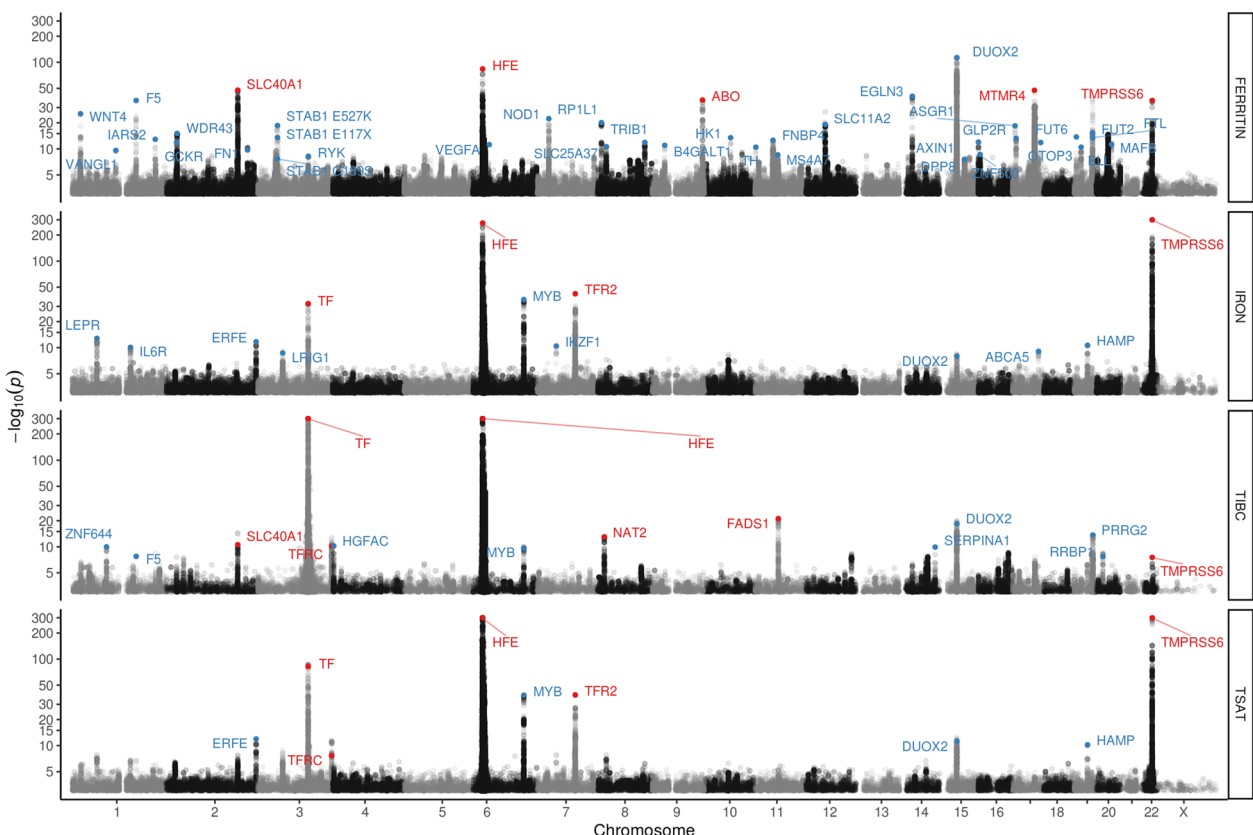

**Fig. 2 Manhattan plots for iron homeostasis biomarker meta-analysis results for ferritin (N = 246,139), serum iron (N = 163,511), total iron-binding capacity (TIBC, N = 135,430), and transferrin saturation (TSAT, N = 131,471).** Variants are plotted by chromosomal position (x-axis) and −log10 P values (y-axis). A likelihood ratio test was used when testing for the association. Blue = novel loci (not reported in previous iron GWAS studies), red = previously reported loci.

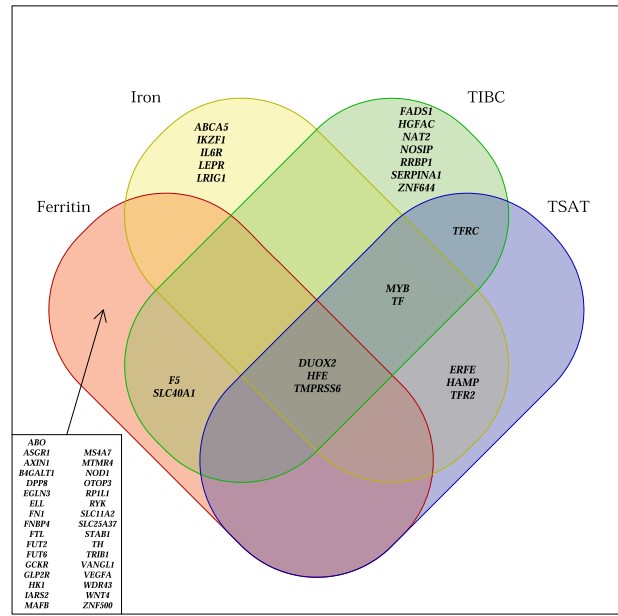

**Fig. 3 Venn diagram.** Venn diagram showing loci (with predicted gene) harboring variants associated with ferritin, iron, TIBC, and/or TSAT.

and liver cells and macrophages[13]. *HAMP*, *HFE*, *TMPRSS6*, *TF*, *TFR2*, and *TFRC* along with the iron storage protein ferritin light chain (encoded by *FTL*) all have variants associated with iron biomarkers. Furthermore, the *MTMR4* variant rs34523089 (MAF = 14.1%), associates with ferritin ($\beta$ = 0.069 SD [0.059,

0.078], $P = 3.2 \times 10^{-48}$). MTMR4 has been shown to localize to early endosomes where it interacts with and dephosphorylates activated R-Smads, thus negatively regulating transforming growth factor $\beta$ (TGF$\beta$) signaling[14] and TGF$\beta$1 has been shown to activate hepcidin mRNA expression[15]. The *MTMR4* variant also associates with hepcidin protein levels in our pQTL study, similar to what was seen with variants in the known hepcidin regulators, *TMPRSS6,* and *HFE* (Supplementary Data 9). The *SERPINA1* p.Glu288Val variant (rs17580[A], MAF = 3.79%) associates with increased TIBC ($\beta$ = 0.076 SD [0.053, 0.099], $P = 1.2 \times 10^{-10}$). *SERPINA1* encodes the protease inhibitor (PI) alpha-1-antitrypsin (A1AT) and the p.Glu288Val variant—also known as the PI S allele—is associated with A1AT-deficiency (A1ATD)[16]. Liver disease in A1ATD has been linked to liver iron overload[17], and recently A1AT was shown to increase hepcidin expression through proteolytic cleavage and inhibition of *TMPRSS6*[18].

*Inflammation*: IL6 and its receptor IL6R are important inflammatory mediators positively regulating liver hepcidin during inflammation[19–21]. The *IL6R* p.Asp358Ala variant (rs2228145[C], MAF = 41%) that associates with decreased risk of rheumatoid arthritis[22,23] associates with an increase in serum iron ($\beta$ = 0.026 SD [0.018, 0.034], $P = 8.4 \times 10^{-11}$). Leptin and its receptor *LEPR*, in addition to its central role as an adipokine, have been shown to control cellular immune responses in several pathological situations including rheumatic diseases[24]. The intergenic variant rs35945185[A] (MAF = 36.5%) linked to *LEPR* associates with iron ($\beta$ = 0.031 SD [0.023, 0.039], $P = 1.54 \times 10^{-13}$). The *IL6R* and *LEPR* associated variants (rs2228145[C], rs35945185[A]) both are negatively associated

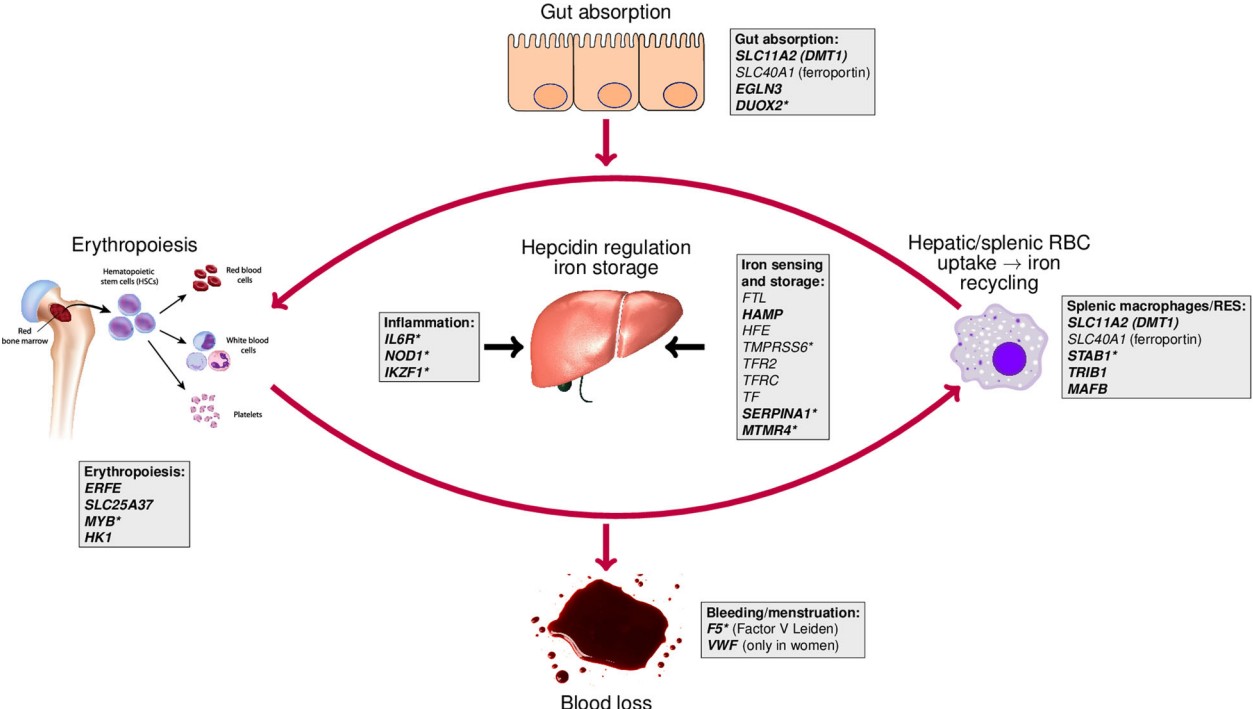

**Fig. 4 Iron homeostasis loci in the context of systemic iron homeostasis, categorization into main physiological processes.** Novel loci are in bold font. *Gene with a predicted probability of being causal (based on a variant-to-gene algorithm, see "Methods") larger than 50%. The liver, blood spot, and erythropoiesis/hematopoiesis cartoons were bought from Shutterstock (standard license), the macrophage is from Wikimedia Commons (https://commons.wikimedia.org/wiki/File:Macrophage.svg).

with inflammatory markers serum amyloid A-1/A-2 proteins, and furthermore, the *LEPR* associated variant is also negatively associated with C-reactive protein (Supplementary Data 9). Also, the rs2529440[T] intron variant (MAF = 45%) at *NOD1*, encoding an intracellular innate immune pattern recognition sensor for bacterial cell components[25], associates with a reduction in ferritin levels ($\beta = -0.035$ SD [$-0.014$, $-0.028$], $P = 4.6 \times 10^{-23}$). Furthermore, the rs12718598[C] intron variant in *IKZF1*, encoding the lymphocyte specification and differentiation transcription factor Ikaros, shown to play a role in auto-immune diseases[26], associates with increased serum iron levels ($\beta = 0.027$ SD [0.019, 0.034], $P = 3.7 \times 10^{-11}$).

*Gut absorption*: Iron absorption is mediated by the two iron transporters DMT1 (encoded by *SLC11A2*) at the luminal side and ferroportin (encoded by *SLC40A1*) basolaterally, both regulated by hepcidin signals and both harboring variants associated with iron homeostasis biomarkers[13]. Recently, hepcidin blocking of intestinal ferroportin was shown to inhibit HIF-2α expression, through increased intracellular iron and subsequent activation of iron-dependent prolyl hydroxylases, leading to reduced expression of iron absorptive proteins[27]. Mammalial HIF-α prolyl hydroxylases are encoded by the three genes *EGLN1-3*[28]. The rs996347[C] intron variant (MAF = 35%) at *EGLN3* associates with increased ferritin ($\beta = 0.049$ SD [0.042, 0.056], $P = 3.0 \times 10^{-41}$). *EGLN3* is a likely candidate to mediate the inhibition of intestinal HIF2α expression, as it specifically inhibits HIF-2α rather than HIF-1α [29,30]. The *DUOX2* p.His678Arg variant (rs57659670[C], MAF = 7.5%) associates with reduced ferritin ($\beta = -0.14$ SD [$-0.16$, $-0.13$], $P = 1.1 \times 10^{-113}$), serum iron ($\beta = -0.042$ SD [$-0.056$, $-0.028$], $P = 1.1 \times 10^{-8}$), and TSAT ($\beta = -0.058$ SD [$-0.074$, $-0.041$], $P = 5.7 \times 10^{-12}$) and increased TIBC ($\beta = 0.077$ SD [0.060, 0.094], $P = 3.7 \times 10^{-19}$). *DUOX2* is expressed in the upper intestinal mucosa and may play a role in innate mucosal immunity[10,31]. Furthermore, in mouse models, *DUOX1* and *DUOX2* knockouts have a greater susceptibility

to *Helicobacter felis* infection and inflammation[32] and epidemiological studies have indicated that *H. pylori* infections in humans are associated with reduced iron stores[33].

*Iron recycling*: recycling of heme–iron takes place in the reticuloendothelial system in the spleen and liver, where old red cells are taken up and iron recycled back to the bone marrow, providing over 90% of the iron needed for the generation of heme in red cell precursors[1]. DMT1 and ferroportin also transport iron from endocytic vesicles and export iron out of the macrophage, respectively[34]. Furthermore, three uncorrelated rare variants (MAF < 1%) in *STAB1* (p.Glu117Ter/rs762752083[T], p.Gly189-Ser/rs750717575[A] and p.Glu527Lys/rs745795585[A]) and a variant in LD with a *STAB1* variant (*GNL3* p.Ser451Thr/rs34216132[C], $r^2 > 0.99$ with the *STAB1* variant p.Ser1089Gly/rs41292856[G]) (Supplementary Fig. 4) all associate with increased ferritin, with effects ranging from 0.17 to 0.35 SD ($P = 2.2 \times 10^{-8}$ to $2.6 \times 10^{-19}$). *STAB1* is primarily expressed in M2-macrophages and sinusoidal endothelial cells[35] and has been shown to affect phosphatidylserine-mediated uptake of aged red blood cells[36,37]. We also report associations of the intergenic variants rs2954029[T] (MAF = 48%) and rs6029148[A] (MAF = 7.1%) with reduced and increased ferritin (rs2954029[T]: $\beta = -0.024$ SD [$-0.031$, $-0.018$], $P = 1.4 \times 10^{-12}$; rs6029148[A]: $\beta = 0.046$ SD [0.033, 0.058], $P = 5.6 \times 10^{-12}$). Their closest protein-coding genes, *TRIB1* (for rs2954029) and *MAFB* (for rs6029148), have both been shown to control the differentiation of macrophages[38,39].

*Erythropoiesis*: The bone marrow relays signals inhibiting liver hepcidin synthesis under a state of stress erythropoiesis to make iron available to erythroid precursors[40]. Variants located close to two known iron regulators within the erythropoiesis compartment, the intergenic variant rs13253974[A] (MAF = 32%) near *SLC25A37* (mitoferrin-1)[41] and the intron variant rs13007705[T] at *ERFE* (erythroferrone)[40,42] associate with increased ferritin

(rs13253974[A]: $\beta = 0.024$ SD [0.017, 0.032], $P = 2.5 \times 10^{-11}$), serum iron (rs13007705[T]: $\beta = 0.029$ SD [0.021, 0.037], $P = 2.0 \times 10^{-12}$) and TSAT (rs13007705[T]: $\beta = 0.033$ SD [0.024, 0.042], $P = 1.1 \times 10^{-12}$). In addition, the rs9399136[C] variant in the *HBS1L-MYB* intergenic region associates with increased serum iron ($\beta = 0.057$ SD [0.049, 0.066], $P = 1.1 \times 10^{-36}$) and TSAT ($\beta = 0.067$ SD [0.057, 0.077], $P = 5.3 \times 10^{-39}$) and reduced TIBC ($\beta = -0.033$ SD [$-0.044$, $-0.023$], $P = 2.7 \times 10^{-10}$), while the rs17476364[C] intron variant in *HK1* associates with increased ferritin ($\beta = 0.043$ SD [0.032, 0.054], $P = 3.6 \times 10^{-14}$). Variants in the *HBS1L-MYB* intergenic region are known to associate strongly with fetal hemoglobin levels[43,44]. Fetal hemoglobin levels are induced during stress erythropoiesis[45,46], a condition also involving ERFE signaling[40]. Furthermore, *HK1* mutations are associated with reduced red cell survival[47].

*Bleeding/menstruation*: Loss of iron occurs primarily through epithelial desquamation and blood loss[48]. Two iron homeostasis variants show sexual dimorphism: The *F5* p.Arg534Gln variant (rs6025[T], MAF = 2.8%; also known as factor V Leiden) associates with increased ferritin ($\beta = 0.15$ SD [0.12, 0.17], $P = 6.9 \times 10^{-37}$) and reduced TIBC ($\beta = -0.093$ SD [$-0.13$, $-0.061$], $P = 1.8 \times 10^{-08}$) and has a six times stronger effect on ferritin in females than in males, while the *VWF* p.Tyr1584Cys variant (rs1800386[C], MAF = 0.94%) associates with reduced ferritin in females only ($\beta = -0.17$ SD [$-0.23$, $-0.12$], $P = 3.0 \times 10^{-10}$). The factor V Leiden variant is associated with a hypercoagulable state[49] and the *VWF* p.Arg534Gln variant is associated with type 2 VWD, a common inherited bleeding disorder[12]. Since both variants show stronger effects in premenopausal than postmenopausal women, and both variants affect clotting, they are likely working through blood loss and primarily menstrual bleeding. In addition, in a meta-analysis using data from Iceland, Denmark, and the UK, the factor V Leiden variant is protective against menorrhagia (OR = 0.82 SD [0.76–0.88], $P = 2.1 \times 10^{-7}$). To further address the effects of variants affecting bleeding and thrombosis we carried out a candidate gene analysis for association with iron homeostasis markers. We screened for coding variants in 375 genes associated with abnormal bleeding (HP:0001892) and 76 genes associated with venous thrombosis (HP:0004936) listed in Human Phenotype Ontology (https://hpo.jax.org/). Nineteen new variants in 14 genes associated with at least one iron homeostasis marker (Supplementary Data 10, Bonferroni corrected *P* value threshold of $1.7 \times 10^{-5}$, ~3000 variants). Only one of these variants in *ARHGAP31* showed sexual dimorphism (effect on ferritin; women ($\beta = 0.230$, $P = 8.4 \times 10^{-8}$), men ($\beta = -0.044$ SD, $P = 0.389$), $P_{het} = 4.0 \times 10^{-5}$) (Supplementary Data 10). One additional missense variant in *IRF2BP2* associated with iron in premenopausal women only (premenopausal women ($\beta = -0.149$ SD, $P = 4.63 \times 10^{-5}$), postmenopausal ($\beta = 0.091$ SD, $P = 0.036$), $P_{het} = 2.40 \times 10^{-5}$) (Supplementary Data 10).

**Iron homeostasis variants and red blood cell traits.** To better understand the effect of the sequence variants on iron homeostasis and iron usage, we tested for association with the red blood cell indices hemoglobin ($N = 286,622$), mean corpuscular hemoglobin ($N = 286,245$), mean corpuscular volume ($N = 286,248$), and reticulocyte count ($N = 19,031$) and compared the effects of variants on them and the four iron biomarkers (Supplementary Fig. 5, Supplementary Data 11). Normally, as body iron stores fall, the hemoglobin concentration, mean corpuscular volume, and mean corpuscular hemoglobin concentration also fall. The p.Cys282Tyr variant at *HFE* (rs1800562) strongly affects all iron and red blood cell biomarkers except reticulocyte count. Variants at *DUOX2*, *F5*, and *TRIB1* have a similar pattern of effects on iron and red blood cell

biomarkers, with a negative effect on TIBC and mainly positive effects on the red cell indices (Supplementary Fig. 5). The variant showing the strongest effect on ferritin is a stop-gained variant in *STAB1* (Stabilin-1) ($\beta = 0.35$ SD [0.26, 0.44], $P = 3.2 \times 10^{-14}$) (Supplementary Data 2). This variant also shows an unusual pattern with decreased hemoglobin along with increased ferritin, indicating that body stores of iron are sufficient but the recycling of iron from stores is abnormally reduced (Supplementary Fig. 5).

**IDA and iron overload.** The two extremes of iron homeostasis, iron deficiency, and iron overload, are clinically important and associated with high disease burden[4,50]. In iron deficiency, depletion of iron stores is followed by reduced iron availability for erythropoiesis, leading to IDA, presenting as hypochromic, microcytic anemia with low ferritin and/or low TSAT[48]. Increased TSAT, most commonly defined as a saturation above 50%, is used as a screening marker for hemochromatosis and iron overload[51]. To understand how the 62 iron homeostasis variants affect either IDA or iron overload, we tested for association with IDA (defined as ever simultaneously having hemoglobin < 120 g/L for women, <130 g/L for men, MCV < 80 fl, MCH < 27 pg and either ferritin < 10 mcg/L or TSAT < 16%; $N_{cases} = 6476$, $N_{controls} = 362,706$)[5] and iron overload (defined as TSAT ever >50%[4], $N_{cases} = 4156$, $N_{controls} = 342,647$) (Fig. 5, Supplementary Data 12), correcting for $2 \times 62 = 124$ performed tests. The missense variants in *DUOX2* (p.His678Arg; rs57659670[C]) and *F5* (p.Arg534Gln, rs6025[T]) associate with IDA (*DUOX2* p.His678Arg: OR = 1.29 [1.20–1.39], $P = 2.0 \times 10^{-11}$; *F5* p.Arg534Gln: OR = 0.60 [0.49–0.73]; $P = 3.4 \times 10^{-7}$). The variants showing sexual dimorphism for the effect on ferritin also showed similar trends with regard to IDA (Supplementary Fig. 6, Supplementary Data 13). In addition, a 3.55 kb deletion in the *SLC11A2* 3′ untranslated region (3′ UTR) and its downstream intron associates with IDA through a recessive mode of inheritance (OR = 32.5 [10.0–105]; $P = 6.4 \times 10^{-9}$) (Fig. 5, Supplementary Data 12, Supplementary Fig. 7). A rare frameshift mutation in *TMPRSS6* (p.Asn473ThrfsTer63, rs773570300) only detected in the Icelandic cohort (MAF = 0.16%) also associated with IDA (OR = 3.0 [2.1–4.4]; $P = 1.2 \times 10^{-8}$). The rs9399136[C] variant in the intergenic *HBS1L/MYB* region is the only variant to associate with both IDA (OR = 0.84 [0.80–0.89], $P = 4.7 \times 10^{-11}$) and iron overload (OR = 1.13 [1.07–1.20], $P = 1.4 \times 10^{-5}$). This variant has not been associated with iron homeostasis but has been associated with hematological traits[52] and variants in the same region have been associated with fetal hemoglobin expression[43,53]. Additionally, variants in the iron homeostasis regulatory genes *HFE*, *TMPRSS6*, *TF*, and *TFR2* associate with iron overload (Fig. 5, Supplementary Data 12).

We tested the 62 iron homeostasis variants for association with the following eleven clinical manifestations of iron overload and/or iron deficiency[54] based on various meta-analyses performed in Iceland using data from Iceland, UK, Denmark, and the USA: hemochromatosis, liver fibrosis/cirrhosis, liver cancer, type 2 diabetes, impotence, cardiomyopathy, osteoporosis, osteoarthritis, hyperpigmentation, amenorrhea, and restless leg syndrome (Supplementary Data 14). Taking all $62 \times 11 = 682$ tests into account using Bonferroni correction, the *TMPRSS6* p.Val749Ala variant (rs855791[A]) associates with less risk of hemochromatosis ($N_{cases} = 719$, $N_{controls} = 497,001$; OR = 0.80 [0.72–0.89], $P = 6.1 \times 10^{-5}$). The *HFE* p.Cys282Tyr variant (rs1800562[A]), the main variant associating with recessive hereditary hemochromatosis (type 1) associates with a higher risk of hemochromatosis (additive model: OR = 25.7 [21.6–30.5], $P < 10^{-300}$; recessive model: OR = 218.9 [164.6–291.0], $P < 10^{-300}$), liver fibrosis/cirrhosis ($N_{cases} = 1043$, $N_{controls} = 705,646$; additive model:

| Phenotype | Gene | Variant | Min/Maj | MAF (%) | OR (95% CI) | P−value |
|---|---|---|---|---|---|---|
| IDA | F5 | rs6025 | T/C | 1.9 | 0.60 (0.49, 0.73) | $3.4 \times 10^{-7}$ |
| IDA | HBS1L/MYB | rs9399136 | C/T | 27 | 0.84 (0.80, 0.89) | $4.7 \times 10^{-11}$ |
| IDA | DUOX2 | rs57659670 | C/T | 8.3 | 1.29 (1.20, 1.39) | $2.0 \times 10^{-11}$ |
| IDA | TMPRSS6 | rs773570300 | T/TG | 0.16 | 3.00 (2.05, 4.37) | $1.2 \times 10^{-8}$ |
| IDA | SLC11A2* | – | [del] | 0.9 | 32.5 (10.0, 105) | $6.4 \times 10^{-9}$ |
| IO | HBS1L/MYB | rs9399136 | C/T | 27 | 1.13 (1.07, 1.20) | $1.4 \times 10^{-5}$ |
| IO | TF | rs748587164 | A/T | 0.12 | 3.89 (2.46, 6.15) | $5.8 \times 10^{-9}$ |
| IO | TF | rs4854760 | G/A | 26 | 0.84 (0.79, 0.89) | $1.0 \times 10^{-8}$ |
| IO | HFE | rs1799945 | G/! | 12 | 1.44 (1.34, 1.55) | $1.3 \times 10^{-24}$ |
| IO | HFE | rs1800562 | A/G | 6.7 | 3.54 (3.29, 3.80) | $2.2 \times 10^{-255}$ |
| IO | TFR2 | rs7385804 | C/A | 35 | 0.90 (0.86, 0.95) | $1.6 \times 10^{-4}$ |
| IO | TMPRSS6 | rs855791 | A/G | 41 | 0.75 (0.71, 0.79) | $1.9 \times 10^{-26}$ |

0.50  1.0  2.0  4.0  8.0  16.0  32.0  64.0
Odds ratio

**Fig. 5 Iron homeostasis variants associated with iron deficiency anemia (IDA) or iron overload (IO).** A forest plot showing the odds ratio (error bars showing 95% confidence intervals) for each of the genetic variants associated with either iron deficiency anemia (IDA) or iron overload (IO). *Results for the SLC11A2 deletion variant are shown for the recessive model while results for other variants are for the additive model. Min/Maj minor/major allele, MAF minor allele frequency, OR odds ratio, CI confidence interval.

OR = 1.50 [1.30–1.74], $P = 4.5 \times 10^{-8}$; recessive model: OR = 5.54 [3.58–8.58], $P = 1.54 \times 10^{-14}$) and liver cancer ($N_{cases} = 844$, $N_{controls} = 792{,}550$; additive model: OR = 1.53 [1.28–1.82], $P = 2.3 \times 10^{-6}$; recessive model: OR = 8.19 [5.04–13.29], $P = 1.8 \times 10^{-17}$), consistent with previous reports[3]. Furthermore, the GCKR p.Leu446Pro variant (rs1260326[T]) associates with a lower risk of type 2 diabetes ($N_{cases} = 36{,}710$, $N_{controls} = 663{,}962$; OR = 0.94 [0.93–0.96], $P = 1.4 \times 10^{-10}$) (Supplementary Data 14). We also generated polygenic risk scores (PRS) for ferritin and TSAT and regressed the scores against the same eleven clinical manifestations of iron overload and/or iron deficiency (Supplementary Data 15). The PRS for ferritin and TSAT only associated with hemochromatosis (ferritin OR = 2.71 [2.49–2.95], $P = 9.4 \times 10^{-119}$; TSAT OR = 3.75 [3.51–4.00], $P < 1 \times 10^{-300}$). Restless leg syndrome has repeatedly been associated with iron deficiency[55,56] and iron supplementation recommended in select cases[57]. We confirm that in the Icelandic restless leg syndrome cohort iron biomarkers suggest increased iron deficiency (lower ferritin ($\beta = -0.07$ SD [−0.12, −0.02] SD; $P = 0.0037$) and TSAT ($\beta = -0.06$ SD [−0.12, −0.001], $P = 0.045$) and higher TIBC ($\beta = 0.14$ SD [0.08, 0.20] SD; $P = 8.5 \times 10^{-6}$) and there is increased incidence of IDA compared to population controls (OR = 1.39 [1.03–1.84], $P = 0.0244$) (Supplementary Data 16). The lack of genetic association seen with either individual iron homeostasis variants or PRS argues against a simple causal relationship between iron deficiency and restless leg syndrome.

**Novel SLC11A2 deletion variant.** Rare loss-of-function mutations in SLC11A2 (solute carrier family 11 member 2 encoding DMT1, divalent metal transporter 1) have been associated with a microcytic anemia with iron overload under the recessive mode of inheritance[58–60] demonstrating a role of DMT1 in both iron absorption and recycling. We identified 14 homozygous carriers of the abovementioned deletion in SLC11A2 in the Icelandic cohort, seven of whom had been diagnosed with IDA (microcytic anemia with low ferritin and/or low TSAT) and one with transfusion-dependent anemia; two had required transfusions and one intravenous iron (Supplementary Data 17).

Transcription of SLC11A2 leads to four major mRNAs with differing tissue-specific expression patterns[61]. These messages differ both in their usage of 5′ exons 1a or 1b and usage of alternative 3′ translated and untranslated regions (UTRs) (Fig. 6A). These alternative UTRs differ in that one contains an iron-response element (IRE), denoted IRE+, while the other UTR lacks such a motif, denoted IRE−. The IRE+ UTR is primarily expressed in duodenal and kidney epithelium, mediates iron absorption, and is regulated directly by cellular iron status through interaction with IRE-binding proteins[62,63]. Of the four highest expressed transcripts in blood, two contain the IRE− UTRs, and two contain the IRE+ UTRs. The SLC11A2 deletion extends from within the IRE+ containing 3′ UTR and into the downstream intron (Fig. 6A). Heterozygotes ($N = 251$) and homozygotes ($N = 2$) express 40% (95% CI: 58–62%, $P = 2.2 \times 10^{-16}$) and 81% (95% CI: 73–87%, $P = 0.015$) less IRE+ transcripts than wildtype ($N = 12{,}828$), respectively (Fig. 6B). When comparing allele-specific transcription in heterozygotes there was a 3.7-fold ($P = 2.2 \times 10^{-17}$) preference for wildtype allele in IRE+ containing alleles. The deletion removes the native 3′ UTR polyadenylation signal, likely resulting in an unstable mRNA. The IRE− transcripts are expressed at 29% greater levels by heterozygotes than noncarriers (95% CI: 26–33%, $P = 2.2 \times 10^{-16}$) and at 133% greater levels by homozygotes (95% CI: 48–205%; $P = 0.016$). These data suggest that the SLC11A2 deletion causes isoform-specific effects, suppressing the expression of IRE+ containing transcripts, that are primarily expressed in absorptive duodenal and kidney epithelium[62] leading to reduced absorption. This leads to a recessive hereditary IDA. Hepcidin levels based on proteomics samples from 35,559 Icelanders are reduced in SLC11A2 deletion carriers ($\beta = -0.172$ SD [−0.257, −0.088], $P = 5.9 \times 10^{-5}$), consistent with systemic iron deficiency[64]. A single homozygous carrier of the deletion has a hepcidin value 2.17 SDs below average. The only previously described genetic IDA, iron refractory IDA, due to homozygous loss-of-function variants in TMPRSS6, is associated with hepcidin dysregulation and inappropriately high hepcidin values[64].

## Discussion

Through a GWAS meta-analysis of the iron homeostasis bio-markers ferritin, serum iron, iron-binding capacity, and TSAT in Iceland, Denmark, and the UK, we have identified 56 loci harboring variants associating with one or more of these biomarkers, 46 of which are novel (including six rare variants, six low-frequency variants, and 37 common variants). Among the novel loci, variants in DUOX2 and SLC11A2 associate with increased risk of IDA, while the F5 rs6025[T] variant protects against IDA. Furthermore, the rs9399136[C] variant at the HBS1L/MYB locus is protective against IDA while increasing the risk of iron overload.

While most of these iron homeostasis variants show similar effects in Iceland, UK, and Denmark, the observed heterogeneity for a subset of the variants may reflect demographic, clinical, and environmental differences. In clinical populations, iron home-ostasis markers are more frequently measured in individuals with

A

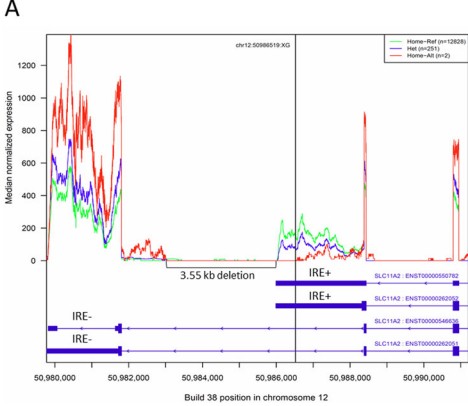

B

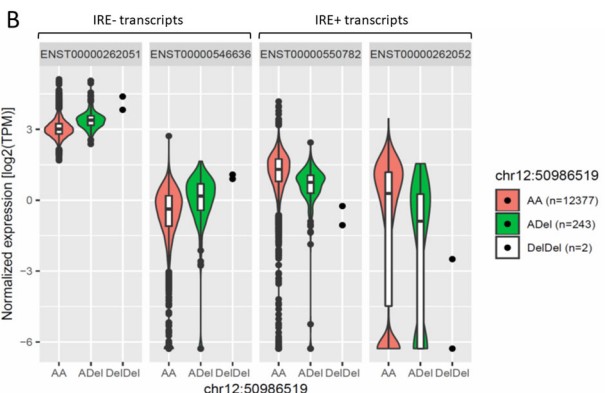

**Fig. 6 Expression levels of *SLC11A2* for novel deletion mutant compared to wild type. A** Coverage plot of RNA-sequenced reads from whole blood showing the median normalized expression at the 3′ UTR end of *SLC11A2* in wild-type (AA, green; *n* = 12,828) heterozygous (ADel, blue; *n* = 251) and homozygous (DelDel, red; *n* = 2) individuals for the different *SLC11A2* deletion genotypes. **B** Comparison of expression levels of the four major SLC11A2 transcripts (two transcripts without iron response elements (IRE) in their 3′ UTR (IRE-) and two with IRE (IRE+) in their 3′ UTR in whole blood RNA using a mixed violin- and boxplot. The violin plot shows the density where the width represents the frequency of the log2 normalized expression levels. The white boxes show the distribution statistics (interquartile range and median) and the whiskers represent ±1.5× the interquartile range. The filled circles correspond to individual expression values or outliers that lie beyond the extremes of the whiskers.

suspected iron deficiency or overload. Blood donors, on the other hand, are screened for anemia and several other diseases at every donation. Therefore, people with a previous history of iron deficiencies are underrepresented in blood donor studies, although a substantial proportion of blood donors will develop low or reduced iron stores[65]. These differences in cohort characteristics could partially explain heterogeneity in effect sizes between populations for a subset of the variants. The sex-specific heterogeneity reported highlights the differences between the sexes in iron homeostasis. The two sequence variants showing the strongest heterogeneity are both variants in coagulation factors, the well-known factor V-Leiden mutation[49] and a mutation in *VWF* known to cause type 2 VWD[12]. These variants are likely to mediate their effects through increased (*VWF*) or decreased (*F5*) blood loss, in women being mainly mediated through menstruation, supported by the finding that factor V Leiden variant protects against menorrhagia. Thirty-four of the 62 reported novel variants only associate with ferritin. Possible explanations of this high number could be that ferritin as a biomarker is affected by a more broad array of processes, such as inflammation

and tissue damage (e.g., liver injury)[66], and also that we have more ferritin measurements (~246 K vs. ~131–163 K).

Iron deficiency is a major global health problem, especially for children and women[2]. A worldwide survey in 2010 showed that one-third of the world population is anemic with iron deficiency being responsible for approximately half of that cases[50]. In addition to the nonspecific symptoms of IDA, it also may contribute globally to reduced cognitive performance in children[67], adverse outcomes of pregnancies[68], and decline in cognition in the elderly[48,69]. Despite the importance of iron deficiency and IDA, no systematic genetic studies looking at iron deficiency or IDA have been performed. Sequence variants that are common (at *DUOX2* and the *HBS1L-MYB* intergenic region), low-frequency (at *F5*), and rare (at *TMPRSS6* and *SLC11A2*) associate with IDA (Fig. 5, Table 1). The association of the missense *DUOX2* variant with all iron homeostasis markers, as well as with IDA is striking. That this association was seen in all three populations studied but not observed in previous GWAS of iron homeostasis is intriguing, however, it should be noted that Benyamin et al.[6] reported a subgenome-wide significant association with ferritin near this locus (rs16976620). Our study is significantly larger and also benefits from more comprehensive imputation panels made available since then, which likely enabled us to not only detect an association at genome-wide significance but also map this to the likely causal gene with high confidence.

The phenotype of recessive IDA with low iron stores that we report with the rare 3.5 kb deletion within *SLC11A2* is different from the previously reported recessive hypochromic anemia with iron overload associated with this gene[55–57]. Further studies to define the pathways mediating the effects of the variants associating with IDA could help shed light on the pathophysiology of iron deficiency. Notably, neither any individual iron homeostasis variants nor the PRS for ferritin or TSAT associate with the risk of restless leg syndrome, a neurological disorder suggested being exacerbated by iron deficiency[70]. Although this argues against a simple causal relationship between the two, a more complex relationship, e.g., through brain iron concentrations[71] cannot be ruled out. Even though hereditary hemochromatosis is most often associated with *HFE* p.Cys282Tyr homozygosity, the penetrance is only 28% in males and much lower in females[72]. The common missense variant in *TMPRSS6* (rs855791[A], MAF = 43.1%) protects against hereditary hemochromatosis (OR = 0.80 [0.72–0.89], $P = 6.1 \times 10^{-5}$) and could thus be a modifying gene in this disease.

In summary, we have identified 46 novel loci affecting iron homeostasis. Many of the novel candidate genes have roles in homeostasis through mechanisms, such as absorption, iron recycling, erythropoiesis, and hepcidin regulation. Furthermore, we show an association of five of these loci with IDA, a major clinical entity that hitherto has not been studied thoroughly from a genetic point of view. This study reveals a substantial catalog of possible iron regulatory genes, awaiting further inquiry to fully elucidate their functional role.

## Methods

**Study subjects from Iceland.** The Icelandic data (where around one-half of all individuals had repeated measurements) include the vast majority of all clinical laboratory results in Iceland from 1990 to 2017. Serum iron and TIBC were measured with colorimetric methods and serum ferritin was measured with an electrochemiluminescence immunoassay using reagents and calibrators and Cobas 6000 and 8000 modular instruments from Roche Diagnostics, Mannheim, Germany. Hemoglobin concentration measurements, as well as other basic hematology parameters used, were measured on EDTA anticoagulated blood using the Sysmex XN-1000 hematology analyzer.

All participants who donated samples gave informed consent and the National Bioethics Committee of Iceland approved the study (VSN-15-198) which was conducted in agreement with conditions issued by the Data Protection Authority of Iceland. Personal identities of the participant's data and biological samples were

encrypted by a third-party system (Identity Protection System), approved, and monitored by the Data Protection Authority.

**Study subjects from the UK**. The INTERVAL study is a prospective cohort study of approximately 45,000 blood donors, representative of the wider donor population, nested in a randomized control trial. Participants, aged 18 years or older, were recruited between 2012 and 2014 from 25 National Health Service Blood and Transplant static donor centers in England. All participants provided written, informed consent, and the study was approved by the Cambridge (East) Research Ethics Committee (ref: 11/EE/0538).

Ferritin measurement was based on the immunological agglutination principle with the enhancement of the reaction by latex. Latex particles coated with anti-ferritin antibodies agglutinated with ferritin and the precipitate was determined turbidimetrically at 570/800 nm. Serum iron was measured using a colorimetric method (FerroZine) without deproteinization. Under acidic conditions, iron was liberated from transferrin. Acrobat reduced Fe3+ to Fe2+ which then reacted with FerroZine to form a colored complex. The color intensity is directly proportional to the iron concentration and was measured photometrically. TIBC was calculated by summing up serum iron and unsaturated iron-binding capacity, which was also measured photometrically. TSAT was calculated by dividing serum iron by TIBC concentration. All data points lying more than 4.5 interquartile range from the median were considered outliers and removed (591 for ferritin, 7 for transferrin, 65 for TSAT, and 37 for serum iron).

The genotyping protocol and quality control procedures for INTERVAL study samples have been described in detail previously[73]. Briefly, DNA extracted from buffy coat was used to assay approximately 820,000 variants and short insertions/ deletions on the Affymetrix Axiom «Biobank» genotyping array (Affymetrix, Santa Clara, California, US). Genotyping was performed in multiple batches of approximately 4800 samples each. Sample QC was performed including exclusions for sex mismatches, low call rates, duplicate samples, extreme heterozygosity, and non-European descent. We carried out high-resolution multiple imputations using a joint UK10K and 1,000 Genomes Phase 3 (May 2013 release) reference panel and retained variants with a MAF ≥ 0.1% and/or INFO score ≥0.4 for analysis.

The meta-analyses of hemochromatosis, liver fibrosis/cirrhosis, liver cancer, type 2 diabetes, osteoarthritis, impotence, cardiomyopathy, osteoporosis, hyperpigmentation, and amenorrhea (Supplementary Data 14) include data from the UK Biobank, accessed under Application Number 56270.

**Study subjects from Denmark**. The Danish Blood Donor Study (DBDS), initiated in 2010 as collaborative blood donor-oriented and generic research platform and is an on-going nation-wide prospective cohort with inclusion sites at all Danish blood collection facilities. Currently, more than 110,000 blood donors are participating, and more than 95% of invited blood donors are willing to participate[74]. Due to the step-wise roll-out of DBDS, an enrichment of individuals from the greater Copenhagen region (the capital) and the central region of Jutland (the second largest city) are present in this study. DBDS has secured necessary permissions and approval from the Danish Data Protection Agency (2007-58-0015) and the Scientific Ethical Committee system (M-20090237). Briefly, regarding the DBDS genomic cohort DNA is purified from whole blood and subsequently stored at −20 degrees Celsius. DBDs participants in this study has been genotyped in 1 batch at Decode genetics using the Global Screening Array by Illumina optimized for comparison with the Illumina Omni Express chip[75]. Ferritin was measured on fresh EDTA-anticoagulated plasma samples using two commercially available assays: for 30,903 individuals using Ortho Vitros 5600 (Ortho Clinical Diagnostics, Rochester, NY, USA), and for 2851 individuals using Abbott Architect i2000SR (Abbott Laboratories, Abbott Park, IL, USA), including 27 individuals that had measurements taken using both methods.

**Whole-genome sequencing**. The process used to whole-genome sequence the 28,075 Icelanders, as well as the subsequent imputation, has been described in recent publications[76,77]. In summary, we sequenced the whole genomes of 28,075 Icelanders using Illumina technology to a mean depth of at least 10× (median 32×). Single-nucleotide polymorphism (SNPs) and indels were identified and their genotypes called using joint calling with Graphtyper[78]. In total, 155,250 Icelanders were genotyped using Illumina SNP chips and their genotypes were phased using long-range phasing[79]. All sequenced individuals were also chip-typed and long-range phased, providing information on haplotype sharing that was subsequently used to improve genotype calls. Genotypes of the 32 million high-quality sequence variants were imputed into all chip-typed Icelanders. Variants in the Icelandic and Danish cohorts were imputed based on the IMPUTE HMM model[80] as previously described[81]. Variants in INTERVAL were imputed using the Sanger Imputation Server (https://imputation.sanger.ac.uk) which implements the Burrows–Wheeler transform imputation algorithm PBWT on whole chromosomes. A combined UK10K and the 1000 Genomes Phase 3 reference panel was used[82]. Using genealogic information, the sequence variants were also imputed into relatives of the chip-typed further increasing the sample size for association analysis and the power to detect associations. All of the variants tested had imputation information over 0.8. The GWAS from Denmark was performed using 19 million markers identified through whole-genome sequencing of 2816 Danes that were

subsequently imputed into 84,386 chip-typed individuals. The GWAS from the UK was performed with 19 million markers from the UK10K and 1000 Genomes Phase 3 reference panel, imputed into 43,059 chip-typed individuals participating in the INTERVAL study. In total, 40 million markers were tested in the meta-analysis.

**Association testing and meta-analysis**. The four iron homeostasis biomarkers (ferritin, serum iron, TIBC, and TSAT) were each rank-based inverse normal transformed to a standard normal distribution (separately for each sex) and adjusted for age using a generalized additive model. In addition, for the UK cohort, the biomarkers were adjusted for menopausal status, ABO blood group, body mass index, smoking levels, alcohol levels, and iron supplementation status. For each sequence variant, the mixed model implemented in the software BOLT-LMM v2.3[83], using the genotype as an additive covariate and the transformed quantitative trait as a response, was used to test for association with quantitative traits. Logistic regression was used to test for association between variants and case-control phenotypes, using software developed at deCODE genetics[76].

We used LD score regression to account for distribution inflation in the dataset due to cryptic relatedness and population stratification[84]. LD score regression intercepts were as follows: ferritin: 1.032 (SE = 0.011), iron: 1.016 (SE = 0.025), TIBC: 1.030 (SE = 0.039), TSAT: 1.025 (SE = 0.020). We used logistic regression to test for association between sequence variants and binary traits, regressing trait status against expected genotype count. In the Icelandic data, we adjusted for sex, age, and county of birth by including these variables in the logistic regression model. In the UK and Danish data we adjusted for sex and age, as well as principal components in order to adjust for population stratification.

Results from the Icelandic, UK, and Danish datasets were combined using a fixed-effect inverse-variance weighted meta-analysis, where different datasets were allowed to have different population frequencies for alleles and genotypes but assumed to have a common effect. Heterogeneity in effect estimates was assessed using a likelihood ratio test. Effects are always given in units of SD. The pooled SD using data from Iceland, UK, and Denmark are 1.08 μg/L for log(ferritin), 7.76 μmol/L for iron, 14.14 μmol/L for TIBC, and 13.25% for TSAT.

We accounted for multiple testing by means of a weighted Bonferroni correction, taking into account the higher prior probability of association of certain variant annotations while controlling the family wise error rate (FWER) at 0.05[9]. The method has been described previously[9] and results in stricter multiple testing correction than the commonly used threshold of $5 \times 10^{-8}$ (which would not control FWER at 0.05 given that 40 million markers were tested) while being more powerful than simply correcting for 40 million tests using a fixed threshold of 0.05/ $40,000,000 = 1.25 \times 10^{-9}$. The resulting significance thresholds were $2.0 \times 10^{-7}$ for high-impact variants (including stop-gained, frameshift, splice-acceptor, or splice-donor variants, $N = 11,723$), $4.0 \times 10^{-8}$ for "moderate-impact" variants (including missense, splice-region variants and in-frame indels, $N = 202,336$), $3.7 \times 10^{-9}$ for "low-impact" variants (including upstream and downstream variants, $N = 2,896,354$) and $6.1 \times 10^{-10}$ or for the "lowest-impact" variants (including intron and intergenic variants, $N = 37,239,641$).

Loci were defined based on physical proximity, where variants in a 500 kb window (lead variant ±250 kb) were considered to be at a single locus. We defined novel loci as loci not reported in previous GWAS of biomarkers of iron homeostasis.

**Variant-to-gene mapping**. To predict the most likely causal gene for each variant we used an algorithm taking into account the gene location with regard to LD class (defined as all variants with $r^2 > 0.8$ with the lead variant), the variant effect for coding variants, and the effect on gene expression (eQTL, restricting to the top cis-eQTL). The algorithm, called variant-to-gene mapping, considers all genes within the LD class ±250 kb and outputs a score for each gene.

Often, the GWAS variant is not causal itself but in LD with the causal variant. To identify the likely causal gene, we defined all variants in linkage disequilibrium ($r^2 > 0.8$) with the GWAS variant as the LD class. We assumed local effects, where genes overlapping the LD class interval receive a distance score of 5, while genes within 250 kb on each side of the LD class interval receive a distance score of 1. The variants in the LD class were then scored based on their capability to affect gene coding (i.e., transcription/translation): a variant with high impact (stop-gain and stop-loss, frameshift indel, donor and acceptor splice-site, and initiator codon variants) was given a coding score of 150, while a variant with moderate impact (missense, in-frame indel, splice region) was given a coding score of 30. For each gene, we summed up the coding scores of all coding variants affecting it, i.e., coding variants within the gene itself. Variants shown to be correlated with gene expression (eQTL) in any tissue received an eQTL score of 50. We restricted ourselves to the top cis-eQTL (lowest P value $< 10^{-7}$, distance from gene < 1 Mb) for each gene and tissue. We assumed that eQTL in different tissues/different variants were due to the same signal. Therefore, we did not sum up the eQTL scores per gene but used the maximum eQTL score per gene. The total score per gene was computed as the maximum of its distance, coding, and eQTL scores. The normalized gene score was computed by scaling such that the sum of normalized scores for all candidate genes was 1, so to enable direct comparisons across genes. Note that this automatically takes the gene density into account. In cases where more than one gene share the maximum score (for example, if the LD class has four genes and they all have probability = 0.25), we chose the gene with the most

significant eQTL if such information existed, otherwise the gene closest to the lead variant was selected. Relative values for the scoring for high- and moderate-impact values were based on enrichment analysis, as previously described[9], while the score of 50 for eQTL was determined in order to make coding and eQTL equally informative overall. Values for proximity were set to have some degree of preference for closeby genes, given otherwise equal evidence, while at the same time giving stronger weight to coding and eQTL than to proximity alone. Data sources for eQTL data are listed in Supplementary Data 18.

**Genetic overlap with other traits**. We calculated genetic correlations between pairs of traits using the cross-trait LD score regression methods[84] in our meta-analysis using summary statistics from traits in the Icelandic and UK datasets. We used results for about 1.2 million variants, well imputed in both datasets and for LD information we used precomputed LD scores for European populations (available from the Broad Institute).

**Heritability estimation**. Heritability was estimated in the following two ways: (1) 2 × parent–offspring correlation, (2) 2 × full sibling correlation, using the Icelandic data (where all family relationships are known).

**Polygenic risk scores**. We generated PRS for ferritin and TSAT and regressed the scores, along with sex, year of birth, and 20 principal components as covariates in logistic regression models against 11 clinical manifestations of iron deficiency or iron overload (restless legs, hemochromatosis, liver fibrosis/cirrhosis, liver cancer, type 2 diabetes, osteoarthritis, impotence, cardiomyopathy, osteoporosis, hyper-pigmentation, and amenorrhea). Scores are based on a framework set of 620,000 high-quality SNPs covering the whole genome, adjusted for LD using LDpred[85]. The methods used to generate the PRS have been previously described in detail[86]. For restless legs, the phenotype data is from Iceland while for the other ten phenotypes, data is from UK Biobank (restless legs were not available in UK Biobank). To minimize bias and/or overfitting, the geographical population with the phenotype data is not included when generating the scores. Thus, for ferritin, the PRS for restless legs is based on a Denmark+UK GWAS meta-analysis, while the PRS for the other phenotypes is based on Iceland + Denmark GWAS meta-analyses. For TSAT, the PRS for restless legs is based on the UK GWAS, while the PRS for the other phenotypes is based on the Icelandic GWAS.

**Protein measurements (pQTL)**. During 2000–2019, we collected plasma samples from 40,004 Icelanders. Fifty-two percent of the samples were collected as part of the Icelandic Cancer Project, while the remaining samples (48%) were collected as part of various genetic programs at deCODE genetics, Reykjavík, Iceland. All samples were measured using the SOMAscan platform, containing 5284 aptamers providing a measurement of relative binding of the plasma sample to each of the aptamers in relative fluorescence units, corresponding to 4,792 proteins. After quality control, unique measurements for $N = 35,559$ individuals were used for genome-wide association analysis.

**Reporting summary**. Further information on research design is available in the Nature Research Reporting Summary linked to this article.

## Data availability
The Icelandic population WGS data have been deposited at the European Variant Archive under accession code PRJEB15197. The authors declare that the data supporting the findings of this study are available within the article, its Supplementary Data files, and upon request. Overall meta-analysis summary statistics have been shared at https://www.decode.com/summarydata/. The UK Biobank data can be obtained upon application (ukbiobank.ac.uk). For this study, UK-Biobank data was under project number 56270.

## Code availability
We used publicly available software (URLs listed below) in conjunction with algorithms described in the Methods section: BWA 0.7.10 mem (https://github.com/lh3/bwa); GenomeAnalysisTKLite 2.3.9 (https://github.com/broadgsa/gatk/); Picard tools 1.117 (https://broadinstitute.github.io/picard/); SAMtools 1.3 (http://samtools.github.io/); Bedtools v2.25.0-76-g5e7c696z (https://github.com/arq5x/bedtools2/); Variant Effect Predictor (https://github.com/Ensembl/ensembl-vep).

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

## Acknowledgements

We thank all the study subjects for their valuable participation, the staff from all studies, and the participating physicians. Participants in the INTERVAL randomized controlled trial were recruited with the active collaboration of NHS Blood and Transplant England (www.nhsbt.nhs.uk), which has supported fieldwork and other elements of the trial. DNA extraction and genotyping were co-funded by the National Institute for Health Research (NIHR), the NIHR BioResource (http://bioresource.nihr.ac.uk/) and the NIHR Cambridge Biomedical Research Centre (BRC) [*]. The academic coordinating centre for INTERVAL was supported by core funding from NIHR Blood and Transplant Research Unit in Donor Health and Genomics (NIHR BTRU-2014-10024), UK Medical Research Council (MR/L003120/1), British Heart Foundation (SP/09/002; RG/13/13/30194; RG/18/13/33946) and the NIHR Cambridge BRC [*]. The views expressed are those of the author(s) and not necessarily those of the NIHR or the Department of Health and Social Care. A complete list of the investigators and contributors to the INTERVAL trial is provided in reference[87]. The academic coordinating centre would like to thank blood donor centre staff and blood donors for participating in the INTERVAL trial. Professor John Danesh holds a British Heart Foundation Professorship and a National Institute for Health Research Senior Investigator Award. Steven Bell and Dragana Vuckovic were funded by the NIHR Blood and Transplant Research Unit in Donor Health and

Genomics (NIHR BTRU-2014-10024). Stephen Kaptoge is funded by a BHF Program Grant (RG/18/13/33946). Will Astle, Joanna Howson, and Tao Jiang are funded by the NIHR Cambridge BRC. Angela M. Wood and Elias Allara are supported by the EC-Innovative Medicines Initiative (BigData@Heart). Praveen Surendran is supported by a Rutherford Fund Fellowship from the Medical Research Council grant MR/S003746/1. This work was supported by Health Data Research UK, which is funded by the UK Medical Research Council, Engineering and Physical Sciences Research Council, Economic and Social Research Council, Department of Health and Social Care (England), Chief Scientist Office of the Scottish Government Health and Social Care Directorates, Health and Social Care Research and Development Division (Welsh Government), Public Health Agency (Northern Ireland), British Heart Foundation and Wellcome. The Novo Nordisk Foundation (NNF14CC0001 and NNF17OC0027594). The Innovative Medicines Initiative 2 Joint Undertaking under grant agreement No. 115881 (RHAP-SODY) (Karina Banasik and Søren Brunak). The Danish Administrative Regions; The Danish Administrative Regions' Bio- and Genome Bank; The authors thank all the blood banks in Denmark for both collecting and contributing data to this study. Danish Blood Donor Research Fund. Aarhus University, Copenhagen University Hospital Research Fund.

## Author contributions

Conception or design of the work: S.B., A.S.R., M.K.M., E.F., U.T., P.S., D.G., K.S., N.S., W.H.O., D.J.R., A.S.B., J.D., E.D.A., and H.U.; the acquisition of data: V.T., B.V., P.T.O., I.O., G.R., O.S., H.S., I.J., H.S., H.H., E.A., A.R., N.S., W.H.O., D.J.R., A.S.B., J.D., E.D.A., A.S.R., E.S., K.S.B., L.W.T., C.E., K.M.D., O.B.V.P., H.P., K.R.N., He.H., T.F.H., D.G.C., K.B., S.B., H.U., and D.B.R.; analysis of data: M.K.M., E.F., G.B., J.K.S., G.H., H.P.E., S.K., S.H.L., K.J., M.I.M., M.L.F., A.O., J.K.S., G.T., D.G., S.B., E.A., and D.W.l.; interpretation of data: M.K.M., E.F., G.T., U.T., P.S., D.G., K.S., S.B., E.A., A.R., D.S.P., J.M.M.H., W.J.A., D.V., Pr.S., T.J., P.A., S.K., A.M.W., J.E.P., N.S., W.H.O., D.J.R., A.S.B., J.D., E.D.A., A.S.R., and H.U; drafting the paper: M.K.M., E.F., P.S., D.G., K.S., S.B., E.A., E.D.A., and A.S.R; and all authors were involved in editing the final paper.

## Competing interests

The authors declare the following competing interests: Henrik Ullum received an unrestricted research grant from Novartis. Cristian Erikstrup received an unrestricted research grant from Abbott. Søren Brunak reports grants from Innovation Fund Denmark, grants from Novo Nordisk Foundation during the conduct of the study; and personal fees from Intomics A/S and Proscion A/S, outside the submitted work. John Danesh sits on the International Cardiovascular and Metabolic Advisory Board for Novartis (since 2010); the Steering Committee of UK Biobank (since 2011); the MRC International Advisory Group (ING) member, London (since 2013); the MRC High Throughput Science 'Omics Panel Member, London (since 2013); the Scientific Advisory Committee for Sanofi (since 2013); the International Cardiovascular and Metabolism Research and Development Portfolio Committee for Novartis; and the Astra Zeneca Genomics Advisory Board (2018). Adam S. Butterworth has received grants outside of this work from AstraZeneca, Biogen, BioMarin, Bioverativ, Merck, and Novartis and personal fees from Novartis. For the authors who are affiliated with deCODE genetics/Amgen, we declare competing financial interests as employees.

## Additional information

[1]The National Institute for Health Research Blood and Transplant Research Unit in Donor Health and Genomics at the University of Cambridge, University of Cambridge, Cambridge, UK. [2]British Heart Foundation Cardiovascular Epidemiology Unit, Department of Public Health and Primary Care, University of Cambridge, Cambridge, UK. [3]Department of Clinical Immunology, Copenhagen University Hospital, Copenhagen, Denmark. [4]deCODE genetics/Amgen Inc., Reykjavik, Iceland. [5]Faculty of Medicine, School of Health Sciences, University of Iceland, Reykjavik, Iceland. [6]Department of Infectious Disease Epidemiology, Faculty of Epidemiology and Population Health, London School of Hygiene and Tropical Medicine, London, UK. [7]Medical Research Council Biostatistics Unit, Cambridge Institute of Public Health, Cambridge, UK. [8]Department of Human Genetics, Wellcome Sanger Institute, Wellcome Trust Genome Campus, Hinxton, UK. [9]Department of Clinical Immunology, Aarhus University Hospital, Aarhus, Denmark. [10]Department of Cardiology, Division Heart & Lungs, University Medical Center Utrecht, Utrecht University, Utrecht, The Netherlands. [11]Rutherford Fund Fellow, Department of Public Health and Primary Care, University of Cambridge, Cambridge, UK. [12]Department of Clinical Immunology, Næstved Hospital, Næstved, Denmark. [13]The Laboratory in Mjodd, RAM, Reykjavik, Iceland. [14]Department of Clinical Immunology, Odense University Hospital, Odense, Denmark. [15]Department of Laboratory Hematology, Landspitali, the National University Hospital of Iceland, Reykjavik, Iceland. [16]Department of Clinical Immunology, Aalborg University Hospital, Aalborg, Denmark. [17]Department of Clinical Biochemistry, Landspitali, the National University Hospital of Iceland, Reykjavik, Iceland. [18]Department of Epidemiology Research, Statens Serum Institut, Copenhagen, Denmark. [19]Danish Headache Center, Department of Neurology, Rigshospitalet-Glostrup, Glostrup, Denmark. [20]Institute of Biological Psychiatry, Copenhagen University Hospital MHC Sct. Hans, Roskilde, Denmark. [21]Novo Nordisk Foundation Center for Protein Research, University of Copenhagen, Copenhagen, Denmark. [22]Department of Clinical Biochemistry, Akureyri Hospital, Akureyri, Iceland. [23]Department of Neurology and Program in Sleep, Emory University School of Medicine, Atlanta, GA, USA. [24]Translational Disease Systems Biology, Novo Nordisk Foundation Center for Protein Research, Faculty of Health and Medical Sciences, University of Copenhagen, Copenhagen, Denmark. [25]Department of Haematology, University of Cambridge, Cambridge, UK. [26]UK National Health Service Blood and Transplant, Cambridge Biomedical Campus, Cambridge, UK. [27]Radcliffe Department of Medicine and National Health Service Blood and Transplant, John Radcliffe Hospital, Oxford, UK. [28]UK National Health Service Blood and Transplant, John Radcliffe Hospital, Oxford OX3 9BQ, UK. [29]School of Engineering and Natural Sciences, University of Iceland, Reykjavik, Iceland. [35]These authors contributed equally: Steven Bell, Andreas S. Rigas, Magnus K. Magnusson, Egil Ferkingstad, Elias Allara. [36]These authors jointly supervised this work: Emanuele Di Angelantonio, Henrik Ullum, Kari Stefansson. *A list of authors and their affiliations appears at the end of the paper. ✉email: magnus.magnusson@decode.is; ed303@medschl.cam.ac.uk; Henrik.Ullum@regionh.dk; kstefans@decode.is

**DBDS Genomic Consortium**

**Denmark** Steffen Andersen[30], Karina Banasik [24], Søren Brunak [24], Kristoffer Burgdorf[3], Christian Erikstrup [9], Thomas F. Hansen [19,20,21], Henrik Hjalgrim[18], Gregor Jemec[31], Poul Jennum[32], Pär Johansson[3], Kasper R. Nielsen[16], Mette Nyegaard[33], Helene M. Paarup [14], Ole B. Pedersen [12], Mikkel Petersen[9], Erik Sørensen[3], Henrik Ullum[3,36 ✉] & Thomas Werge[34]

**Iceland** Daniel F. Gudbjartsson [4,29], Kari Stefansson [4,5,36 ✉], Hreinn Stefánsson[4] & Unnur Thorsteinsdóttir[4,5]

[30]Department of Finance, Copenhagen Business School, Copenhagen, Denmark. [31]Department of Clinical Medicine, Sealand University Hospital, Roskilde, Denmark. [32]Department of Clinical Neurophysiology at University of Copenhagen, Copenhagen, Denmark. [33]Department of Biomedicine, Aarhus University, Aarhus, Denmark. [34]Institute of Biological Psychiatry, Mental Health Centre Sct. Hans, Copenhagen University Hospital, Roskilde, Denmark.

