## [Peer Review File · Communications Biology]

Reviewers' comments:

Reviewer #1 (Remarks to the Author):

In this interesting paper, the authors perform a meta-analysis of iron homeostasis biomarkers. Quite a few of the genes identified have a known role in iron metabolism and hematology. The association of DUOX2 was unexpected and interesting. In their figures, they distinguish between novel and previously reported loci. They could expand the legend to define novel as a mutation of a gene, known or unknown, that has not previously been reported. In figure 6, the figure legend is not sufficiently explanatory. There are ways in which the authors could increase the appeal of this paper for those who are not GWAS experts. For instance, it would be helpful to define MAF upon first use in the ms. They addressed an interesting question by noting that they found no associations with restless legs syndrome. They could help the reader by revealing how many of these subjects had or would be expected to have RLS. It's an interesting approach to gaining insight into this unresolved question about Iron deficiency and restless legs.

There are some minor points:

line 232- associated

lines 320 and 423- hypochromic

refs 76 and 77 are identical

line 162, please explain the significance of the numbers in the heritability of biomarkers- some readers may not know how to interpret this information

Reviewer #2 (Remarks to the Author):

Overall: Thank you for the opportunity to review the manuscript, "A genome-wide meta-analysis yields 46 new loci associating with biomarkers of iron homeostasis". This manuscript describes a genome-wide meta-analysis of several iron-related traits using three well-known cohorts from Iceland, Denmark, and the UK. As a follow-up to locus discovery, the authors also attempted to classify the resulting loci into biologic categories and candidate target genes. My comments are as follows:

Major comments:

- The methods are lacking details with regards to what programs/methods were used for different analyses. For example, what program was used for imputation? What program was used for the association testing? Each program/method for any analysis step has its own set of strengths and limitations and, as such, should be reported by name and/or reference.
- Variant-to-gene mapping algorithm: what is the source of the eQTL data used within this algorithm? If the source is not publicly available (e.g. GTEx), more details are needed with regards to the data used for eQTLs (ex. Sample size, tissues available, source, etc).
- How was a locus defined? How was it determined if a locus was novel or not? (E.g. "...56 loci, of which 46 are novel.")
- Results, lines 62-67: what can you say about the other 8 variants/associations that are not present/shared across the three studies?
- Results, sexual dimorphism: why was this done for the set of 62 variants and not the full set of 87 associations?
- More details are needed with regards to the pQTL analysis. It appears from the Supplemental Table 7 that you only considered the GWAS and pQTL variant to be colocalized/shared when the LD r^2 between the lead GWAS and lead pQTL variant are in $r^2 > 0.80$ (which this reviewer agrees with)- however, this is not stated anywhere in the text or methods.
- The rationale should be stated for why you tested for association of the lead GWAS variants and pQTLs. I assume that the purpose was to help identify candidate target genes?
- Results, line 99-104: "...of the 56 identified loci..." – are you making an assumption that the loci with multiple signals are all acting through the same gene/mechanism? There have been several recent publications that untangle loci with multiple signals to show they are acting on separate

genes. What is the rationale for the assumption?

- Can the overall meta-analysis summary statistics be made publicly available for download through the GWAS catalog, or other source?

Minor comments:

- Results, lines 37-52: are the variants being referred to the lead variants from the association signal? Clarify.
- Supplemental Figure 1: I think the legend on the right is missing the corresponding heat map colors for each of the correlation values.
- What are the available sample sizes when splitting the dataset into men vs women and pre- vs post-menopausal women? Are the differences in pre- vs. post- menopausal women observed because of a much larger sample of pre-menopausal women? Or biologically plausible?

Reviewer #3 (Remarks to the Author):

Very interesting paper! I think some mostly modest changes to the presentation of results will make the many novel and interesting findings clearer to the reader, especially those who have not read much previous literature on iron hemostasis and biology.

The DUOX2 coding variant is quite interesting, I'm a bit surprised it hasn't been discovered in previous papers. Do you think this is due to poor imputation quality, lack of sample size, population differences in allele frequency, etc? Could be discussed a bit more, given the relatively large effect size and consistent association with all 4 iron biomarkers. It is also very interesting to see the female specific VWF and FV Leiden association. Did you do any look-ups of other hemostasis related Mendelian variants (or top variants from GWAS of VWF and other hemostasis factors) to see if there were additional (potentially sub-genome wide significant) results? Enrichment analysis? I think this link to Mendelian bleeding phenotypes and menstruation in pre-menopausal women is underexplored in previous iron GWAS and very interesting/clinically relevant.

What is the overall % pre-menopausal women? In fact, I don't see a cohort characteristics table, which I would like to see. For SNPs with a significant p-value for heterogeneity in Supplementary Table 5 (different men vs women), does this difference persist comparing only men and post-menopausal women?

Why are so many variants associated with ferritin only? Confounding because of ferritin's strong association with inflammation more broadly (signals are not specific to iron related pathways)? Or sample size (looks like Denmark study contributes ferritin only)? How many of these additional loci would still be genome-wide significant without additional samples from Denmark (those in particular seem to me to need to be explained... why don't they effect any other iron index if the sample size available is similar)? Do you see evidence of ferritin associated loci having less clear links to iron metabolism? Was there any pattern of reduced probability of association with red cell indices in Supplemental Table 8 for "ferritin only" variants? This needs to be discussed more. I notice you drop these loci from Table 1 and display them in Table 2 instead, which suggests to me the authors think they are likely to be distinct and potentially less related to iron metabolism too. The LD class concept needs to be explained more clearly in the main text. What reference population was used to calculate LD patterns, which of course differs even among European ancestry populations? How many loci have no confident gene mapping? How many are mapped just based on physical proximity of the lead SNPs or LD proxies versus eQTL or other evidence? What sources do you use for the eQTL data? I don't find this part terribly clear from current wording in the main text or methods. Is eQTL data and/or coding/splicing related evidence necessary for "high confidence" gene assignment?

Some of the literature review in the results section to my mind is more appropriate for the discussion. I would suggest condensing the results section quite a bit, moving content to the

discussion, and organizing a based on categories in Figure 4... lots of interesting things here, but stronger organization by "theme" (erythropoiesis vs iron storage vs blood loss) is needed in the results and discussion (use Figure 4 more, it's really nice!). For the loci that aren't mapped to a category, is this mostly because mapping of variants to genes is unclear, or the biology of the linked genes (as related to iron) is unclear? This should be better clarified when you introduce Figure 4.

I don't have a ton of confidence in the 8 associations (out of the 87) that are found in a single cohort only (though some, like the Icelandic rare frameshift mutation in *TMPRSS6*, are clearly credible), without further replication. Are most of these associations even included in the main results tables? Everything but the *STAB1* variants seems to have a test of heterogeneity, which I assume isn't possible if there's only one cohort included... apologies if I'm missing something! Have you considered a polygenic risk score approach for looking at your variants with clinical outcomes? I think this would be a helpful addition to Supplementary Table 11.

Minor comments: Make sure to define abbreviations in the figure legends, such as iron-response element (IRE) in Figure 6.

Small issue with duplicate references:

76. Bulik-Sullivan, B.K., et al., LD Score regression distinguishes confounding from polygenicity in genome-wide association studies. *Nature Genetics*, 2015. 47(3): p. 291-+.

77. Bulik-Sullivan, B.K., et al., LD Score regression distinguishes confounding from polygenicity in genome-wide association studies. *Nat Genet*, 2015. 47(3): p. 291-5.

Reviewer #4 (Remarks to the Author):

The authors conducted a meta-analysis of four iron homeostasis traits (ferritin, serum iron, iron binding capacity and transferrin saturation) across three European studies. They replicate all previous iron homeostasis GWAS loci, and have heavy emphasis on extensive annotation and curation via the literature, with additional analysis on other iron biomarkers, women's health, iron deficient anemia vs. overload, and some clinical data follow-up. They include a variant-to-gene mapping approach involving scoring with information from LD, eQTLs, pQTLs, and coding variants.

The authors have identified dozens of novel associations, and the complete replication of previous loci lends credence to the phenotype data quality. These new loci require eventual replication or functional validation, and will serve as candidate genes for further work by other analysts and experimentalists in the field.

Comments:

1. Could the LDSC intercepts be included for these GWAS, for a better sense of inflation and polygenicity?
2. Variant-to-gene mapping: how is an "essential" variant defined? Could more details be provided in the text as to how the actual values for the scoring were determined?
3. Some minor typos: "assicated" -> associated (line 232); $62 \times 11 = 683$ -> 682 (line 347)

#	Referee	Response to reviewer
	Referee #1: iron metabolism	
	General comment: In this interesting paper, the authors perform a meta-analysis of iron homeostasis biomarkers. Quite a few of the genes identified have a known role in iron metabolism and hematology. The association of DUOX2 was unexpected and interesting.	We thank the reviewer for their positive comments and interest in the findings presented in our manuscript. We are also grateful for their critical feedback which we will respond to in a point-by-point manner below.
	Major comments	
1	In their figures, they distinguish between novel and previously reported loci. They could expand the legend to define novel as a mutation of a gene, known or unknown, that has not previously been reported.	In our paper we have defined novel loci as genetic loci that have not been reported in previous GWAS studies of iron homeostasis. We have now stated this more clearly in results (page 6, lines 34-36): “We found associations with iron homeostasis biomarkers represented by 62 sequence variants at 56 loci, of which 46 have not been reported in previous GWAS on iron homeostasis and are therefore considered novel” As the reviewer suggests, we have also added this information to the legend of Figure 2 (page 35): “Blue = novel loci (not reported in previous iron GWAS studies), red = previously reported loci.”
2	In figure 6, the figure legend is not sufficiently explanatory.	We have now expanded the legend for Figure 6. Pages 39-40: “Figure 6: A) Coverage plot of RNA-sequenced reads from whole blood showing the median normalized expression at the 3’UTR end of SLC11A2 in wild-type (AA, green; n = 12,828) heterozygous (ADel, blue; n = 251) and homozygous (DelDel, red; n = 2) individuals for the different SLC11A2 deletion genotypes. B) Comparison of expression levels of the four major SLC11A2 transcripts (two transcripts without iron response elements (IRE) in their 3’ UTR (IRE-) and two with IRE (IRE+) in their 3’ UTR in whole blood RNA using a mixed violin- and boxplot. The violin plot shows the density where the width represents frequency of the log2 normalized expression levels. The white boxes shows the distribution statistics (interquartile range and median) and the whiskers represent ±1.5× the interquartile range. The filled circles correspond to individual expression values or outliers that lie beyond the extremes of the whiskers.”

3 There are ways in which the authors could increase the appeal of this paper for those who are not GWAS experts. For instance, it would be helpful to define MAF upon first use in the ms.

We thank the reviewer for this suggestion and have corrected this omission by stating “minor allele frequency (MAF)” in full the first instance we use the term in the results section (Results, page 6, line 54)

4	They addressed an interesting question by noting that they found no associations with restless legs syndrome. They could help the reader by revealing how many of these subject had or would be expected to have RLS. It's an interesting approach to gaining insight into this unresolved question about Iron deficiency and restless legs.	We have now added additional data. In the Icelandic data, iron deficiency anemia cases have a higher probability of also having RLS than the population controls (OR=1.39, 95% CI: 1.03-1.84, P=0.02). Also, RLS cases have lower ferritin (Effect: -0.07 SD, P=0.003) and transferrin saturation (Effect: -0.06 SD, P=0.04) and higher TIBC (0.1351 SD P=8.51E-06) than the population controls. We have also added polygenic risk scores for ferritin and TF saturation (see reviewer #3, comment 8). The polygenic risk score based on a meta-analysis of the Danish and UK data shows no association with RLS and as stated in our previous version, none of the iron homeostasis variants show association with RLS. We believe this argues against a simple causal relationship between iron deficiency and RLS and have added this to our results and discussion. We have added further text to the results and discussion, as follows: Results (page 16, lines 321-333: “We also generated polygenic risk scores (PRS) for ferritin and transferrin saturation, and regressed the scores against the same eleven clinical manifestations of iron overload and/or iron deficiency (Supplementary Table 15). The PRS only associated with hemochromatosis (ferritin OR=2.71 (2.49, 2.95), p= 9.4 x 10⁻¹¹⁹, TSAT OR=3.75 (3.51, 4.00) p<1 x 10⁻³⁰⁰). Restless leg syndrome has repeatedly been associated with iron deficiency (PMID: 8085504, 9646381) and iron supplementation recommended in select cases (PMID: 29425576). We confirm that in the Icelandic restless leg syndrome cohort iron biomarkers suggest increased iron deficiency (lower ferritin (β=-0.07 [-0.12, -0.02] SD; P= 0.0037) and TSAT (β=-0.06 [-0.12, -0.001] SD; P= 0.045) and higher TIBC (β=0.14 [0.08, 0.20] SD; P=8.5 x 10⁻⁶) and there is increased incidence of iron deficiency anemia compared to population controls (OR=1.39 [1.03-1.84]; P=0.0244) (Supplementary table 16). The lack of genetic association seen with either individual iron homeostasis variants or PRS argues against a simple causal relationship between iron deficiency and restless leg syndrome.” Discussion (page 19-20, lines 415-422: “Notably, neither any individual iron homeostasis variants nor the PRS for ferritin or TSAT associate with risk of restless leg syndrome, a neurological disorder suggested to be exacerbated by iron deficiency [66]. Although this argues against a simple causal relationship between the two, a more complex relationship, e.g. through brain iron concentrations (ref: PMID: 25231526) cannot be excluded.”
	Minor comments	
1	line 232- associated	Corrected (page 10, line 199)

2	lines 320 and 423- hypochromic	Corrected (Page 14, line 303 & Page 14, line 303)
3	refs 76 and 77 are identical	We apologise for this duplication; Reference 77 has now been removed
	line 162, please explain the significance of the numbers in the heritability of biomarkers- some readers may not know how to interpret this information	We have added further explanation to the text: Results (page 7, lines 72-75): " Furthermore, we estimated the heritability of iron homeostasis biomarkers to be between 0.16 and 0.32 using parent-offspring and sibling correlations, suggesting that heritability explains 16-32% of the variance of the four iron homeostasis markers studied (Supplementary Table 5). "
Referee #2: GWAS, cardiometabolic traits		
	General comment: Thank you for the opportunity to review the manuscript, "A genome-wide meta-analysis yields 46 new loci associating with biomarkers of iron homeostasis". This manuscript describes a genome-wide meta-analysis of several iron-related traits using three well-known cohorts from Iceland, Denmark, and the UK. As a follow-up to locus discovery, the authors also attempted to classify the resulting loci into biologic categories and candidate target genes. My comments are as follows:	Many thanks for reviewing and commenting our manuscript.
Major comments		
1	The methods are lacking details with regards to what programs/methods were used for different analyses. Each program/method for any analysis step has its own set of strengths and limitations and, as such, should be reported by name and/or reference.	We have added further information to the Methods section and have also included a section titled "Code availability" (page 31) listing the publicly available software that was used; these are expanded upon in turn below.

a	What program was used for imputation?	Variants in the Icelandic and Danish cohorts were imputed using software developed at deCODE genetics based on the IMPUTE HMM model, where chip genotyped individuals that share haplotypes with individuals in the set of sequenced individuals (the training set) are imputed to have the alleles on the background of the shared haplotypes. Variants in INTERVAL cohort were imputed using the Sanger Imputation Server (https://imputation.sanger.ac.uk) which implements the Burrows-Wheeler transform imputation algorithm PBWT on whole chromosomes as described here below. The Methods section has been updated to clarify this: Methods (page 26 , lines 557-561): “Variants in the Icelandic and Danish cohorts were imputed using software developed at deCODE genetics based on the IMPUTE HMM model [PMID: 17572673] as previously described [PMID: 25977816]. Variants in INTERVAL were imputed using the Sanger Imputation Server (https://imputation.sanger.ac.uk) which implements the Burrows-Wheeler transform imputation algorithm PBWT on whole chromosomes. A combined UK10K and the 1000 Genomes Phase 3 reference panel was used [PMID: 26368830].”
b	What program was used for the association testing?	For quantitative traits, association testing was performed using a linear mixed model implemented by BOLT-LMM [PMID: 25642633]. For case-control data, association testing was performed using logistic regression using software developed at deCODE Genetics. We have updated the Methods section as follows: Methods (page 26, lines 576-580): “For each sequence variant, the mixed model implemented in the software BOLT-LMM [PMID: 25642633], using the genotype as an additive covariate and the transformed quantitative trait as a response, was used to test for association with quantitative traits. Logistic regression was used to test for association between variants and case-control phenotypes, using software developed at deCODE genetics [PMID: 25807286].”
2	Variant-to-gene mapping algorithm: what is the source of the eQTL data used within this algorithm? If the source is not publicly available (e.g. GTEx), more details are needed with regards to the data used for eQTLs (ex. Sample size, tissues available, source, etc).	The eQTL data is based on both Icelandic data (not publicly available) and publicly available data. A new Supplementary Table 18 describes the data sources with links to the source publications/databases, sample sizes and available tissues. The text in Methods has also been changed, as follows (page 29, lines 646-647): “Data sources for eQTL data are listed in Supplementary Table 18”

3	How was a locus defined? How was it determined if a locus was novel or not? (E.g. "...56 loci, of which 46 are novel.")	To address this question we have added the following text to the Methods section (page 28, lines 607-609, end of the "Association testing and meta-analysis" subsection): "Loci were defined based on physical proximity, where variants in a 500 kb window (lead variant +- 250 kb) were considered to be at a single locus. We defined novel loci as loci not reported in previous GWAS of biomarkers of iron homeostasis." See also the response to reviewer #1, comment #1.
4	Results, lines 62-67: what can you say about the other 8 variants/associations that are not present/shared across the three studies.	The 8 associations are with five variants at three different loci and they are all rare Icelandic variants. These variants are specific to Iceland and we can therefore not attempt to replicate their association. Even though these associations lack replication we feel that they are all likely to be genuine regulators of iron homeostasis as explained in newly included text specifically addressing this: Results (pages 7-8, line 81-90): "Eight associations are reported with five rare variants at three loci found only in Iceland (MAF=0.12-0.47%): three coding variants (2 missense, 1 stop-gained) in STAB1, 1 stop gained variant in TF, and 1 stop gained variant in TMPRSS6. Common variants associating with iron biomarkers are reported in all three populations for each of these loci, providing additional evidence for these associations (Supplementary Table 2)."
5	Results, sexual dimorphism: why was this done for the set of 62 variants and not the full set of 87 associations?	We have run this analysis for the full set of 87 associations, see results shown in Supplementary Table 6. Note that in our analysis there are 62 independent variants that have 87 associations (some variants have associations with more than one biomarker as shown in the Venn diagram, Figure 3). We have also added one more sentence on the sex and pre/postmenopausal effects in the results: Results (page 8, line 101-104: "In addition we find sex differences in variants in the well-known iron regulatory genes HFE (ferritin, iron and TSAT) and TMPRSS6 (iron) (Supplementary Table 6), again with stronger effects in pre- vs postmenopausal women (Supplementary Table 7)."
6	More details are need with regards to the pQTL analysis. It appears from the Supplemental Table 7 that you only considered the GWAS and pQTL variant to be colocalized/shared when the LD r2 between the lead GWAS and lead pQTL variant are in r2>0.80 (which this reviewer	The reviewer is correct - colocalization is defined as $r^2 > 0.80$. This has now been clarified. Results, page 8-9, lines 113-114 "Among the 62 variants, 30 have at least one associated pQTL, where we use $r^2 > 0.8$ as the limit for considering variants as associated."

	agrees with)-however, this is not stated anywhere in the text or methods.	
7	The rationale should be stated for why you tested for association of the lead GWAS variants and pQTLs. I assume that the purpose was to help identify candidate target genes?	The rationale is to gain further insight into possible biological pathways involved in iron homeostasis. We have changed the text in the following way: Results (page 8, lines 109-112) “To gain further insight into the biological pathways involved in iron homeostasis, we tested for association of the 62 iron homeostasis variants (including all variants with $r^2 \geq 0.8$ with any iron homeostasis variants) with expression of 4,792 proteins in serum using the Somalogic Somascan platform based on samples from 35,559 Icelanders.”
8	Results, line 99-104: “...of the 56 identified loci...” – are you making an assumption that the loci with multiple signals are all acting through the same gene/mechanism? There have been several recent publications that untangle loci with multiple signals to show they are acting on separate genes. What is the rationale for the assumption?	It should be noted that within the novel loci there is only one locus with multiple signals reported, i.e. STAB1 . STAB1 has four coding variants, three missense and one stop-gained. In this case we are presuming one gene. With regard to selection of candidate genes we have added a more detailed description of this process in the results section to further clarify our selection method. Results, (page 6, lines 40-50): “A variant-to-gene mapping algorithm that takes into account gene location, variant effect (for coding variants) and effect on gene expression (eQTL) for each variant (lead variant and LD class) was used to choose a single candidate gene for each locus (see Supplementary Methods). 25 of the 62 iron homeostasis associated sequence variants have a high-confidence predicted causal gene, 23 variants have multiple top-scoring genes, 36 variants have at least one coding variant or eQTL in the LD class, and 13 variants have more than one gene with coding variants and/or eQTL in the LD class (Supplementary Table 3). The LD class of a variant is defined as all variants having $r^2 > 0.8$ with the variant. Linkage disequilibrium (r^2) is estimated based on the Icelandic population. In cases where variants had more than one top-scoring gene, the gene closest to the lead variant was selected, except for two loci where likely candidate genes were present among the top scoring genes (FTL (ferritin light chain) and HAMP (hepcidin)) (Supplementary Table 3).”
9	Can the overall meta-analysis summary statistics be made publicly available for download through the GWAS catalog, or other source?	Overall meta-analysis summary statistics will be made available at the point of publication at https://www.decode.com/summarydata/ . The “Data availability” section in Methods has been updated to reflect this as follows (page 31, lines 701-702): “Overall meta-analysis summary statistics have been shared at https://www.decode.com/summarydata/”
	Minor comments	

1	Results, lines 39-43: are the variants being referred to the lead variants from the association signal?	For the variant-to-gene mapping algorithm analysis we consider the lead variant (and LD class, i.e. variants having $r^2 > 0.8$ with the lead variant) for all 62 identified sequence variants associating with iron homeostasis. This has been clarified in the text: Results (page 6, lines 40-42: “A variant-to-gene mapping algorithm that takes into account gene location, variant effect (for coding variants) and effect on gene expression (eQTL) for each variant (lead variant and LD class) was used to choose a single candidate gene for each locus (see Supplementary Methods)”
2	Supplemental Figure 1: I think the legend on the right is missing the corresponding heat map colors for each of the correlation values.	We have checked carefully supplemental figure 1 and are not able to see any missing correlation values. We hope the reviewer is now able to see corresponding correlation values.
3	What are the available sample sizes when splitting the dataset into men vs women and pre- vs post-menopausal women? Are the differences in pre- vs. post- menopausal women observed because of a much larger sample of pre-menopausal women? Or biologically plausible?	We have added a Supplementary Table 1 detailing summary data for the three cohorts, showing the relevant biomarker and blood count results split into various subgroups, sex, pre- and postmenopausal. We have measurements on 84,328 premenopausal and 49,631 postmenopausal women in our cohorts. The differences between the groups are seen in the effect sizes (beta) and given the large sample size we believe these are biologically plausible differences.
Referee #3: GWAS		
	General comment: Very interesting paper! I think some mostly modest changes to the presentation of results will make the many novel and interesting findings clearer to the reader, especially those who have not read much previous literature on iron hemostasis and biology.	We thank the reviewer for their constructive response to our study and its findings.
Major comments		
1	The DUOX2 coding variant is quite interesting, I'm a bit surprised it hasn't been discovered in previous papers. Do you think this is due to poor imputation quality, lack of sample size,	We agree that this finding is important and that it is surprising that it is not reported in previous studies. The paper by Benyamin et al (Nature Comm) did report a non-genome wide significant association in this locus on Chr15 using a gene based analysis, but did not point to DUOX2 as a candidate gene. We have now added a brief discussion on this topic

	population differences in allele frequency, etc? Could be discussed a bit more, given the relatively large effect size and consistent association with all 4 iron biomarkers.	Discussion (page 19, lines 403-410): “The association of the missense DUOX2 variant with all iron homeostasis markers, as well as with IDA is striking. That this association was seen in all three populations studied but not observed in previous GWAS of iron homeostasis is intriguing, however, it should be noted that Benyamin et al. reported a sub-genome wide significant association with ferritin near this locus (rs16976620) (PMID: 25352340). Our study is significantly larger and also benefits from more comprehensive imputation panels made available since then, which likely enabled us to not only detect an association at genome-wide significance but also map this to the likely causal gene with high confidence.”
2	It is also very interesting to see the female specific VWF and FV Leiden association. Did you do any look-ups of other hemostasis related Mendelian variants (or top variants from GWAS of VWF and other hemostasis factors) to see if there were additional (potentially sub-genome wide significant) results? Enrichment analysis? I think this link to Mendelian bleeding phenotypes and menstruation in premenopausal women is underexplored in previous iron GWAS and very interesting/clinically relevant.	We agree that the female dominant effect seen with VWF and FV is very interesting. We also thank the reviewer for their suggestion and have added a candidate gene analysis focusing on coding variants in all abnormal bleeding (375 genes) and venous thrombosis genes (76 genes) listed in the Human Phenotype Ontology database (https://hpo.jax.org/). Using a Bonferroni corrected p-value threshold of 1.7×10^{-5} (~3000 variants) we found an association in 19 new variants in 14 genes with at least one iron homeostasis marker (see new Supplement Table 10). Only 1 of these variants in ARHGAP31 showed sexual dimorphism (effect on ferritin; women (0.230 SD, $P=8.4 \times 10^{-8}$), men (-0.044 SD, $P=0.389$), $P_{\text{het}}=4.0 \times 10^{-5}$). One additional missense variant in IRF2BP2 showed association with iron in premenopausal women only (premenopausal women (-0.149 SD, $P=4.63 \times 10^{-5}$), postmenopausal (0.091 SD, $P=0.036$), $P_{\text{het}}=2.40 \times 10^{-5}$). We have added supplementary table 10 and text in the results section describing these findings: Results (page 13 lines 239-249): “ To further address the effects of variants affecting bleeding and thrombosis we carried out a candidate gene analysis for association with iron homeostasis markers. We screened for coding variants in 375 genes associated with abnormal bleeding (HP:0001892) and 76 genes associated with venous thrombosis (HP:0004936) listed in Human Phenotype Ontology (https://hpo.jax.org/). 19 new variants in 14 genes associated with at least one iron homeostasis marker (Supplementary Table 10, Bonferroni corrected p value threshold of 1.7×10^{-5}, ~3000 variants). Only one of these variants in ARHGAP31 showed sexual dimorphism (effect on ferritin; women ($\beta=0.230$, $P=8.4 \times 10^{-8}$), men ($\beta=-0.044$ SD, $P=0.389$), $P_{\text{het}}=4.0 \times 10^{-5}$)

		(Supplementary Table 10). One additional missense variant in IRF2BP2 associated with iron in premenopausal women only (premenopausal women ($\beta=-0.149$ SD, $P=4.63 \times 10^{-5}$), postmenopausal ($\beta=0.091$ SD, $P=0.036$), $P_{het}=2.40 \times 10^{-5}$) (Supplementary Table 10). ”
3	What is the overall % pre-menopausal women? In fact, I don't see a cohort characteristics table, which I would like to see. For SNPs with a significant p-value for heterogeneity in Supplementary Table 5 (different men vs women), does this difference persist comparing only men and post-menopausal women?	Among the women that can be reliably classified as either pre- or post-menopausal, the percentage of pre-menopausal women are 59% for those with TIBC measurements, 59% for those with TSAT measurements, 60% for those with serum iron measurements and 63% for those with ferritin measurements (61% pre-menopausal women overall). We have added Supplementary Table 1 showing a cohort characteristics table with sample sizes and summary statistics broken down by sex and menopausal status. The variants at F5, SLC25A37, DOUX2 and VWF that show a greater effect in women than men do not show difference in effect comparing only men and post-menopausal women (see new Supp.Table 8 containing the comparison). However, the difference persists when comparing with post-menopausal women for the variants at HK1 and HFE (which have stronger effects in men). The following text has been added Results (page 8, lines 104-106): “For the variants at F5, SLC25A37, DOUX2 and VWF that show a greater effect in women, the difference does not persist when comparing only men and post-menopausal women (Supplementary Table 8).”
4	Why are so many variants associated with ferritin only? Confounding because of ferritin's strong association with inflammation more broadly (signals are not specific to iron related pathways)? Or sample size (looks like Denmark study contributes ferritin only)? How many of these additional loci would still be genome-wide significant without additional samples from Denmark (those in particular seem to me to need to be explained... why don't they effect any other iron index if the sample size available is similar)? Do you see evidence of ferritin associated loci having less clear links to iron metabolism? Was there any pattern of reduced probability of association with red cell indices in	We do not have a conclusive explanation for the number of variants that only associate with ferritin. We agree that ferritin as a biomarker is likely to be affected by more processes, such as inflammation broadly and tissue damage. We also see that 4 variants that have broad, nonspecific effects on many proteins (ABO, FUT2, GCKR and ASGR1, each with over 50 pQTL associations) only associate with ferritin. We also have greater statistical power because of higher sample numbers compared to the other iron biomarkers ($n= \sim 246K$ vs $131-163K$). The reason we report the ferritin only variants in table 2 (not in table 1) was only because of table size limitations, not importance. We have added the following text to the discussion (page 18-19, lines 391-394): “34 of the 62 reported novel variants only associate with ferritin. Possible explanations of this high number could be that ferritin as a biomarker is affected by more processes such as inflammation and tissue damage (e.g. liver injury) (PMID: 24549403) and that we have more ferritin measurements ($\sim 246K$ vs $\sim 131-163K$).”

	Supplemental Table 8 for “ferritin only” variants? This needs to be discussed more. I notice you drop these loci from Table 1 and display them in Table 2 instead, which suggests to me the authors think they are likely to be distinct and potentially less related to iron metabolism too.	
5	The LD class concept needs to be explained more clearly in the main text. What reference population was used to calculate LD patterns, which of course differs even among European ancestry populations? How many loci have no confident gene mapping? How many are mapped just based on physical proximity of the lead SNPs or LD proxies versus eQTL or other evidence? What sources do you use for the eQTL data? I don’t find this part terribly clear from current wording in the main text or methods. Is eQTL data and/or coding/splicing related evidence necessary for “high confidence” gene assignment?	The LD class for a variant v contains all variants that have $r^2 > 0.8$ with v. This has now been clarified at the first mention of the term “LD class”. Results (page 6, lines 44-47) “...36 variants have at least one coding variant or eQTL in the LD class, and 13 variants have more than one gene with coding variants and/or eQTL in the LD class (Supplementary Table 3). The LD class of a variant is defined as all variants having $r^2 > 0.8$ with the variant. Linkage disequilibrium (r^2) is estimated based on the Icelandic population.” The Icelandic population was used to calculate LD patterns. 29 of the 56 identified loci do not have any variants with a high-confidence gene mapping (defined as having no gene with probability > 0.5 of being the causal gene based on the variant-gene-algorithm). Of the variants that have a high-confidence gene mapping, all except one are based on eQTL or coding evidence (the exception being IARS2 which is selected based on proximity alone). Full information on the evidence behind the gene assignment is available in Supplementary Table 2 (containing the basis for the scores and reasoning behind gene selection) and Supplementary Table 18 (containing the eQTL data sources). We have changed the text in the results section to clarify this (Results, page 6, lines 40-50): “Variant-to-gene mapping algorithm that takes into account gene location, variant effect (for coding variants) and effect on gene expression (eQTL) for each variant (lead variant and LD class) was used to choose a single candidate gene for each locus (see Supplementary Methods). 25 of the 62 sequence iron homeostasis associated variants have a high-confidence predicted causal gene, 23 variants have multiple top-scoring genes, 36 variants have at least one coding variant or eQTL in the LD class, and 13 variants have more than one gene with coding variants and/or eQTL in the LD class (Supplementary Table 3). The LD class of a variant is defined as all variants having $r^2 > 0.8$ with the variant. Linkage disequilibrium (r^2) is estimated based on the Icelandic

		population. In cases where variants had more than one top-scoring gene, the gene closest to the lead variant was selected, except for two loci where likely candidate genes were present among the top scoring genes (FTL (ferritin light chain) and HAMP (hepcidin)) (Supplementary Table 3).”
6	Some of the literature review in the results section to my mind is more appropriate for the discussion. I would suggest condensing the results section quite a bit, moving content to the discussion, and organizing a based on categories in Figure 4... lots of interesting things here, but stronger organization by “theme” (erythropoiesis vs iron storage vs blood loss) is needed in the results and discussion (use Figure 4 more, it’s really nice!). For the loci that aren’t mapped to a category, is this mostly because mapping of variants to genes is unclear, or the biology of the linked genes (as related to iron) is unclear? This should be better clarified when you introduce Figure 4.	We understand that this chapter of the results section is long. On the other hand we feel that it is difficult to split this section between the results and discussion. We would like to keep this section as one and feel that it is most appropriate as a results chapter but if requested we could move it to the discussion. We have also added subheadings to each paragraph to hopefully make it clearer.
7	I don’t have a ton of confidence in the 8 associations (out of the 87) that are found in a single cohort only (though some, like the Icelandic rare frameshift mutation in TMPRSS6, are clearly credible), without further replication. Are most of these associations even included in the main results tables? Everything but the STAB1 variants seems to have a test of heterogeneity, which I assume isn’t possible if there’s only one cohort included... apologies if I’m missing something!	As explained above (Reviewer #2, comment #4) the 8 associations are with five variants at three different loci and they are all rare Icelandic variants. These variants are specific to Iceland and thus not replicated. Even though these associations lack replication we feel that they are all likely to be true regulators of iron homeostasis for the reasons described in the new text. See pages 7-8, line 81-90: “Eight associations are reported with five rare variants at three loci found only in Iceland (MAF=0.12-0.47%): three coding variants (2 missense, 1 stop-gained) in STAB1, 1 stop gained variant in TF, and 1 stop gained variant in TMPRSS6. Common variants associating with iron biomarkers are reported in all three populations for each of these loci, providing additional evidence for these associations (Supplementary Table 2).”
8	Have you considered a polygenic risk score	We have calculated polygenic risk scores (PRS) for ferritin and transferrin saturation

	approach for looking at your variants with clinical outcomes? I think this would be a helpful addition to Supplementary Table 11.	(TSAT), which are the iron biomarkers that are thought to be most directly relevant for clinical outcomes, and assessed the association of these PRS with the eleven clinical outcomes in our original Supplementary Table 11. The results can be found in the new Supplementary table 15. The only phenotype with any evidence of association with the PRS is with hemochromatosis, where a 1 SD increase in the ferritin PRS gives an odds ratio (OR) of 2.71 (95% CI: 2.49-2.95; P=9.35E-119) and a 1 SD increase in the TSAT PRS gives OR=3.75 (3.51-4.00), P<1E-300). The following text has been added Results (p.16, lines. 321-325): “We also generated polygenic risk scores (PRS) for ferritin and transferrin saturation, and regressed the scores against the same eleven clinical manifestations of iron overload and/or iron deficiency (Supplementary Table 15). The PRS for ferritin and TSAT only associated with hemochromatosis (ferritin OR=2.71 (2.49, 2.95), p= 9.4 x 10⁻¹¹⁹, TSAT OR=3.75 (3.51, 4.00) p <1 x 10⁻³⁰⁰).” We have also added a methods description of PRS. Methods, pages 30-31, lines 664-677.
Minor comments		
1	Make sure to define abbreviations in the figure legends, such as iron-response element (IRE) in Figure 6.	This has now been corrected.
2	Small issue with duplicate references: 76. Bulik-Sullivan, B.K., et al., LD Score regression distinguishes confounding from polygenicity in genome-wide association studies. Nature Genetics, 2015. 47(3): p. 291-+. 77. Bulik-Sullivan, B.K., et al., LD Score regression distinguishes confounding from polygenicity in genome-wide association studies. Nat Genet, 2015. 47(3): p. 291-5.	We apologise for this replication; Reference # 76 has now been removed.
Referee #4:		
	General comment: The authors conducted a meta-analysis of four iron homeostasis traits (ferritin, serum iron, iron binding capacity and	We thank the reviewer for their positive comments and recognising the clinical and scientific significance of our work.

	transferrin saturation) across three European studies. They replicate all previous iron homeostasis GWAS loci, and have heavy emphasis on extensive annotation and curation via the literature, with additional analysis on other iron biomarkers, women's health, iron deficient anemia vs. overload, and some clinical data follow-up. They include a variant-to-gene mapping approach involving scoring with information from LD, eQTLs, pQTLs, and coding variants. The authors have identified dozens of novel associations, and the complete replication of previous loci lends credence to the phenotype data quality. These new loci require eventual replication or functional validation, and will serve as candidate genes for further work by other analysts and experimentalists in the field.	
Major comments		
1	Could the LDSC intercepts be included for these GWAS, for a better sense of inflation and polygenicity?	We have added this information to the Methods section (page 27, lines 584-585, after the sentence starting with "We used LD score regression..."): "LD score regression intercepts were as follows: Ferritin: 1.032 (SE=0.011), Iron: 1.016 (SE=0.025), TIBC: 1.030 (SE=0.039), TSAT: 1.025 (SE=0.020)."
2	Variant-to-gene mapping: how is an "essential" variant defined? Could more details be provided in the text as to how the actual values for the scoring were determined?	The high-impact category (giving a coding score of 150) consists of loss-of-function variants (stop-gain and stop-loss, frameshift indel, donor and acceptor splice-site and initiator codon variants). The text has been updated to clarify this, and the reference to "essential" variants has been removed. The updated text reads: Methods: (page 28, lines 622-625) "A variant with high impact (stop-gain and stop-loss, frameshift indel, donor and acceptor splice-site and initiator codon variants) was given a coding score of 150, while a variant with moderate impact (missense, in-frame indel, splice region) was given a coding score of 30."

		Regarding the values for the scoring, the scores for moderate-impact (score: 30) and high-impact variants (score: 150) are based on the enrichments from Sveinbjornsson et al (2016) [PMID: 26854916]. The score for eQTLs (score: 50) was chosen to make eQTL approximately equally important to coding variants overall. Scores for proximity (score 5 for genes overlapping LD class interval, score 1 for genes with 250 kb of LD class interval) were chosen in order to have some preference for closeby genes, while assuming that this evidence is less compelling than that from eQTL or coding variants. While we admit that there is some arbitrariness in the choice of these scores, their main function is to provide a ranking of the possible genes: the exact values of the scores for the various genes is not that important, what is more interesting is the ranking of genes implied by the scoring. To clarify these points, we have added the following to the Methods section (page 29, lines 642-646): “Relative values for the scoring for high- and moderate-impact values were based on enrichment analysis, as previously described [9], while the score 50 for eQTL was determined in order to make coding and eQTL equally informative overall. Values for proximity were set to have some degree of preference for closeby genes, given otherwise equal evidence, while at the same time giving stronger weight to coding and eQTL than to proximity alone.”
	Minor comments	
	"assicated" -> associated (line 232)	Corrected (page 10, line 199)
	62×11=683 -> 682 (line 347)	Thank you for picking this up, also corrected.

REVIEWERS' COMMENTS:

Reviewer #2 (Remarks to the Author):

Thank you for satisfactorily responding to my comments and concerns. I have no additional comments.

Reviewer #3 (Remarks to the Author):

The authors have been very responsive to my comments, and I think the paper looks good. I liked the additions of the PRS and the look-ups for other bleeding related genes. Nice work!

Reviewer #4 (Remarks to the Author):

The authors have sufficiently addressed all my questions in their responses and in the text.